# Polarization lidar for detecting dust orientation: System design and calibration.

Alexandra Tsekeri[1], Vassilis Amiridis[1], Alexandros Louridas[2], George Georgoussis[2],
Volker Freudenthaler[3], Spiros Metallinos[1], George Doxastakis[2], Josef Gasteiger[4], Nikolaos Siomos[1],
Peristera Paschou[1], Thanasis Georgiou[1], George Tsaknakis[2], Christos Evangelatos[2], and
Ioannis Binietoglou[5]

[1]Institute for Astronomy, Astrophysics, Space Applications and Remote Sensing, National Observatory of Athens, Athens, Greece
[2]Raymetrics S.A., Athens, Greece
[3]Fakultät für Physik, Meteorologisches Institut, Ludwig-Maximilians-Universität, Munich, Germany
[4]University of Vienna, Faculty of Physics, Vienna, Austria
[5]National Institute of R&D for Optoelectronics, Magurele, Romania

**Correspondence:** Alexandra Tsekeri (atsekeri@noa.gr)

**Abstract.** Dust orientation is an ongoing investigation in recent years. Its potential proof will be a paradigm shift for dust remote sensing, invalidating the currently used simplifications of randomly-oriented particles. Vertically-resolved measurements of dust orientation can be acquired with a polarization lidar designed to target the off-diagonal elements of the backscatter matrix which are non-zero only when the particles are oriented. Building on previous studies, we constructed a lidar system emitting

linearly- and elliptically-polarized light at 1064 nm and detecting the linear and circular polarization of the backscattered light. Its measurements provide direct flags of dust orientation, as well as more detailed information of the particle microphysics. The system also employs the capability to acquire measurements at varying viewing angles. Moreover, in order to achieve good signal-to-noise-ratio in short measurement times the system is equipped with two laser sources emitting in interleaved fashion, and two telescopes for detecting the backscattered light from both lasers. Herein we provide a description of the optical and

mechanical parts of this new lidar system, the scientific and technical objectives of its design, and the calibration methodologies tailored for the measurements of oriented dust particles. We also provide the first, preliminary, measurements of the system during a dust-free day. The work presented does not include the detection of oriented dust (or other oriented particles), and therefore the instrument has not been tested fully in this objective.

## 1   Introduction

Mineral dust is one of the most important aerosol types in terms of mass and optical depth (e.g., Tegen et al., 1997) and therefore significantly impacts radiation (e.g., Li et al., 2004), while it also interacts with liquid or ice clouds modifying their optical properties and lifetimes (e.g., DeMott et al., 2003), affecting in addition precipitation processes (e.g., Creamean et al., 2013). Once dust particles are deposited at the surface of the Earth they provide micro nutrients to the ocean (e.g., Jickells et al. , 2005) or to land ecosystems (e.g., Okin et al., 2004). For these reasons, IPCC (2013) identified mineral dust and its

impacts on weather, climate and biogeochemistry, along with the associated uncertainties in climate projections, as key topics for future research.

      All aforementioned impacts are strongly affected by the dust size. Observations show that the coarse mode of dust (Weinzierl et al., 2017; Ryder et al., 2018), or even of giant dust particles (van der Does et al., 2018), can be sustained during long-range transport. The process is simulated in dust transport models, which, however, consistently overestimate large particle

removal, as repeatedly found through comparisons of model simulations against measurements (e.g., Adebiyi and Kok, 2021). This discrepancy between observations and theory suggests that other processes counterbalance the effect of gravity along transport. A possible explanation is the electrification of the particles in the dust plumes (Nicoll et al., 2011; Harrison et al., 2018; Daskalopoulou et al., 2021a) and the retainement of the larger dust particles due to the subsequent electric force (Mallios et al., 2020; Toth III et al., 2020).

A side-effect of dust electrification may be the preferential orientation of the nonspherical dust particles (Ulanowski et al., 2007; Mallios et al., 2021). If present, particle orientation will play a role in the radiation reaching the surface of the Earth and the top of the atmosphere, as well as in the interpretation of the remote sensing observations used for dust monitoring from space, that cannot be described using the long-established assumption of randomly-oriented particles. The phenomenon of dust orientation has been extensively studied for space dust (e.g., Whitney and Wolff, 2002), whereas the investigation for

the Earth's atmosphere is a relatively new field of research. Specifically, the only indication of dust orientation in the Earth's atmosphere comes from astronomical polarimetry measurements of dichroic extinction during a dust event at the Canary islands (Ulanowski et al., 2007) and a dust event at the Antikythera island in Greece (Daskalopoulou et al., 2021b). However these measurements refer to column-integrated values, not being capable for vertically-resolved retrievals.

      Lidars (light detection and ranging) are capable of providing vertically-resolved measurements of dust orientation in the

atmosphere. Previous studies have demonstrated the feasibility of using circularly- or linearly-polarized lidar measurements to detect the orientation of ice crystals in clouds (Kaul et al., 2004; Hayman et al., 2012; Balin et al., 2013; Volkov et al., 2015; Kokhanenko et al., 2020) and it has been theoretically shown that these techniques can be extended to oriented dust particles (Geier and Arienti, 2014). Specifically, Geier and Arienti (2014) demonstrated that although the linearly-polarized measurements provided by most of the lidar systems are sufficient for discerning ice crystal orientation, this is not the case

for smaller particles as dust, for which we expect much lower differences than the order(s) of magnitude reported for oriented particles in clouds. What they suggested is to use light that is linearly-polarized along a plane at an angle $\neq 0$, or circularly-polarized light, and detect the backscattered light at different polarization planes. With this approach they showed that the off-diagonal elements of the backscatter phase matrix could be retrieved, providing information on the orientation of the particles.

In this work we propose a different approach for the polarization lidar we designed and constructed, aiming for direct measurements of dust orientation, without having to retrieve the individual off-diagonal elements of the backscatter phase matrix, and also aiming at increasing the information content of the measurements for dust microphysical properties. The new lidar, nicknamed WALL-E, employs two lasers, with the first emitting linearly-polarized light along a plane at $45^o$ with respect to the horizon, and the second emitting elliptically-polarized light, with the angle of the polarization ellipse at $5.6^o$

with respect to the horizon and degree of linear polarization of 0.866. The two laser sources emit interleavingly and their backscattered light is collected also interleavingly, for both, by two telescopes. At the detection side, after the first telescope, the linear polarization of the backscattered light is measured, whereas after the second telescope the circular polarization of the backscattered light is measured. The operating wavelength is at 1064 nm for better probing the dust coarse mode. The system is capable of measuring at more than one zenith and azimuth viewing angles so as to provide more information on

the dust orientation and microphysical properties, depending on the angle of the particle orientation. In order to derive the orientation properties of the particles with respect to the horizon, we define the polarization of the emitted and detected light with respect to the horizon. To achieve highly-accurate polarization measurements with high signal-to-noise ratio (SNR), the lidar system uses high-power lasers (i.e., Class 4 lasers), large aperture telescopes and small receiver field-of-view. For the design and calibration of the system we followed the high quality standards of the European Aerosol Research Lidar Network

(EARLINET) (Freudenthaler, 2016).

  In Section 2 we present a general description of the instrument, focusing on the optical and the mechanical parts of the system. In Section 3 we describe the methodology we followed for the system design, based on specific scientific and technical objectives. In Sections 4 and 5 we present the various calibration procedures. In Section 6 we provide a methodology for comparing the measurements with the ones from commonly-used polarization lidars, using the volume linear depolarization

ratio. In Section 7 we present the first measurements of the system, acquired during a dust-free case in Athens, Greece (we should note that the instrument has not acquired measurements of oriented particles yet). In Section 8 we provide an overview of this work and we discuss its future perspectives. The detailed calculations for the methodologies presented herein are provided in the Appendices A, B and C, as well as in the Supplement. Moreover, a table containing all acronyms and symbols is also provided in the Supplement.

## 2 Overview of the lidar components and operation

The lidar system is equipped with two emission units, and two detection units. The lasers in the emission units emit interleaved light pulses, and their backscattered light is measured interleavingly for each laser at the detectors of the detection units. Each of the detection units is comprised of a telescope, polarizing optics and two detectors. The system uses this "two-laser/two-telescope/four-detector" setup to record eight separate signals with good SNR in short measurement times.

The lidar has been developed by Raymetrics S.A. and it is housed in a compact enclosure that permits the system to perform measurements in the field, under a wide range of ambient conditions. As shown in Fig. 1, the system is comprised by the upper "head" part, containing the lasers, the telescopes and the detection units of the system, the bottom "electronics compartment", containing the power supplies of the lasers, the Transient Recorders (TRs), the Master Trigger Control Unit and the Lidar Peripheral Controlling unit (LPC), and the "positioner" which holds the head and facilitates the measurements at various viewing

angles, which are set manually by changing the azimuth and zenith position of the head. The electronics compartment and the head are connected with two umbilical tubes that contain the cooling lines of the lasers and the power and communication

cables. Figure 2 shows a sketch of the emission and detection units of the system, with a detail description provided in Section 3.

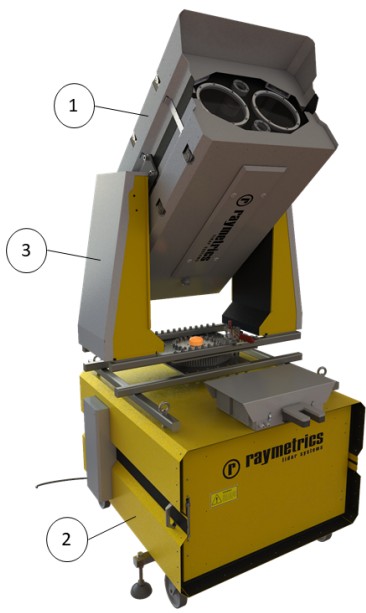

**Figure 1.** The lidar system, with the "head" part at the top (1), the "electronics compartment" at the bottom (2), and the "positioner" of the head (3).

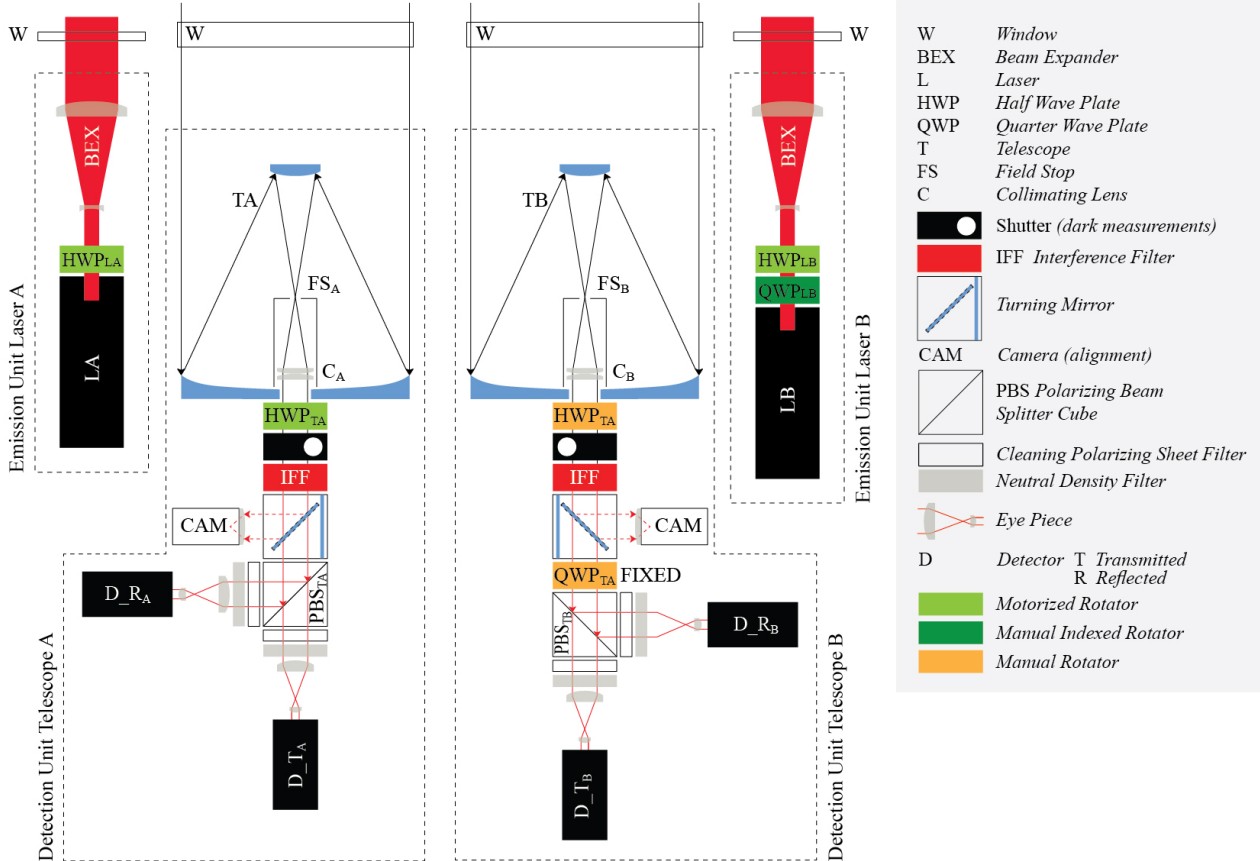

**Figure 2.** Sketch of the emission and detection units of the system: two lasers shooting alternatively ($LA$ and $LB$), with the backscattered signals correspondingly alternatively collected by two telescopes ($TA$ and $TB$) and then redirected at two detectors for each telescope ($D\_s$, $s = T, R$ as of "Transmitted" and "Reflected" channels). As described in detail in in Section 3, the polarization of the light emitted from each laser is changed appropriately, using the $HWP_{LA}$ for laser A and the $QWP_{LB}$ followed by the $HWP_{LB}$ for laser B. The laser beam of each laser is expanded with a beam expander (BEX). At the detection unit after the first telescope the light goes through the $PBS_{TA}$ and after the second telescope the light goes through $QWP_{TB}$ and $PBS_{TB}$. The $HWP_{TA}$ at telescope A is used to correct the rotation of the $PBS_{TA}$ (Section 4.1). The $HWP_{TB}$ at telescope B is used to check the position of the $QWP_{TB}$ with respect to the $PBS_{TB}$ (Section S4 in Supplement). The shutter at each telescope is used for performing dark measurements, and the interference filter is used for reducing the background light in the measurements. The camera at each telescope is used for the alignment of the laser beams with the field-of-view of the telescope.

Due to the analog operation at 1064 nm, the time range of the measurements is restricted by the dark signal changes, which are mainly affected by the change of the (internal) system temperature. The investigation of the acceptable temperature changes, and corresponding acceptable time ranges during which the dark signal does not change considerably, is a work in

progress, with first results to set the acceptable temperature changes to $\pm 2$ °C, which require a new dark measurement every 0.5 hour during summertime, or every 2 hours during wintertime. Also, due to the high power of the lasers there is no eye safety classification for the lidar, although the beam is expanded 5 times. This restricts the operation of the system when there

are no aircrafts at the airspace of the measurements.

## 2.1 The head

The head of the system contains the emission and detection units (Fig. 2), with the lasers at the first, and the telescopes, the polarizing optical elements and the detectors at the latter. The lasers and telescopes are placed in a diamond-shaped layout (Fig. 3, right), that ensures equal distances of both lasers from both telescopes, for the proper alignment of the laser beams with the

field-of-view of both telescopes.

We use Nd:Yag lasers (CFR400 from Lumibird S.A.), emitting at 1064 nm, with energy per pulse of $\sim 250$ mJ (Table 1). We expand the laser beams by 5 times with beam expanders of Galilean type. Each laser and beam expander are mounted on a metallic plate which ensures their alignment and proper expanding of the outgoing laser beam. In front of the lasers we place appropriate optical elements in order to change the polarization state of the emitted light, as described in Section 3. Specifically,

in front of the "laser A" we place a Half Wave Plate ($HWP$) to change the plane of its linear polarization to the plane at $45^o$ with respect to the horizon, and in front of the "laser B" we place a Quarter Wave Plate ($QWP$) followed by a $HWP$, to change its linear polarization to elliptical polarization, with the angle of the polarization ellipse at $5.6^o$ with respect to the horizon, and degree of linear polarization of 0.866. The $QWP$ and the $HWPs$ are mounted on stepping-index rotational mounts (with accuracy of $0.1^o$), which enable us to accurately rotate them to different positions and produce the desired polarization states,

as described in Section 4.

The telescopes are of Dall-Kirkham type, with an aperture of 200 mm, focal length of 1000 mm (F#5), field stop with diameter of 2 mm and a field of view of 2 mrads. The full overlap of the laser beams to the telescope field-of-views is achieved above ranges of 400-600 m.

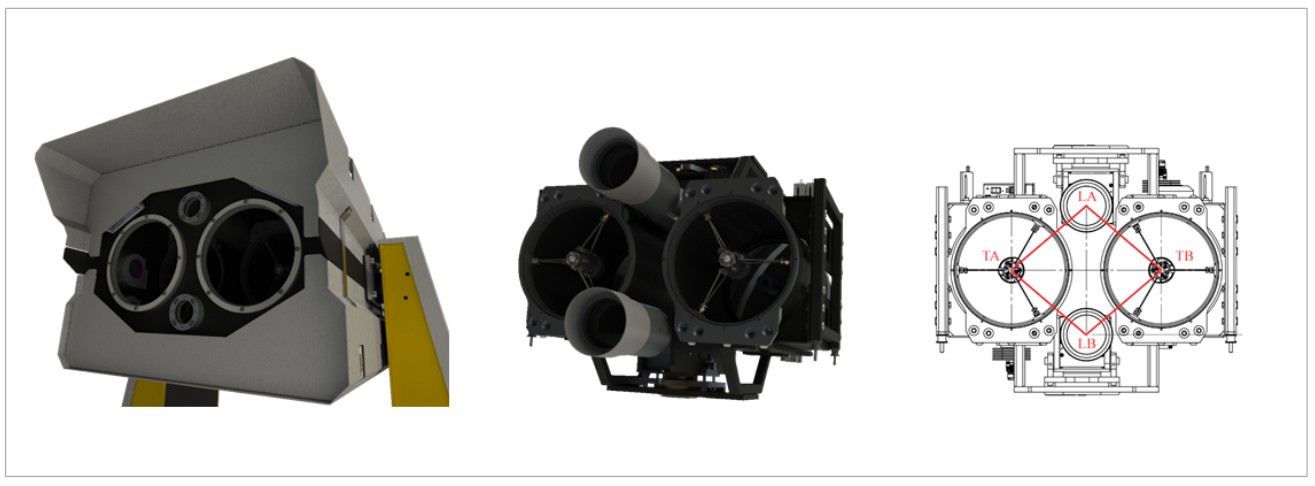

**Figure 3.** The lidar head. Left: Cover of the lidar head, showing the windows in front of the two lasers and the two telescopes. Middle: Photorealistic image of the internal parts of the head, showing the lasers and their beam expanders, the telescopes and the rest of the detection units. Right: Front view of the head, showing the diamond-shaped layout of the lasers (LA and LB) and telescopes (TA and TB).

**Table 1.** Laser A and B parameters

|  | laser A | laser B |
| --- | --- | --- |
| Nominal wavelength (nm) | 1064 | 1064 |
| Energy (mJ) | 256.1 | 256.9 |
| Near Field Beam Diameter (mm) | 6.25 | 6.14 |
| Pulse Width - FWHM (nsec) | 7.29 | 7.46 |
| Divergence at 86.5 % (mrad) | 1.11 | 0.73 |
| Pulse Rate (Hz) | 20 | 20 |

The detection units after the telescopes (Fig. 2) contain the optical elements (e.g. $HWP$, $QWP$, Polarizing Beam Splitter Cube ($PBS$)) that alter the Stokes vector of the collected backscattered light, so as to measure its polarization state effectively, as discussed in detail in Section 3. The optical elements are well-aligned with each other, considering the high tolerance for misalignment due to the emitting divergence at $0.2$ mrad and the field of view of 2 mrad. The signals are recorded by two cooled Avalanche PhotoDiodes (APDs) at each detection unit, which contain remote-controlled power supplies and cooling units. We operate the APDs in Analog mode (not geiger). The signals from the APDs are pre-amplified and digitized by a 16

bit A/D with a sampling rate of 40 MHz and bandwidth of DC to 20 MHz. After digitization, the signals are stored as mVolts at the hard disk of the embedded computer. Each detection unit contains also a shutter for performing dark measurements, a camera for the alignment of the lasers with the telescopes, and an interference filter for reducing the background light in the measurements (Fig. 2). Figure 4 provides a photo-realistic depiction for the detection unit after telescope A.

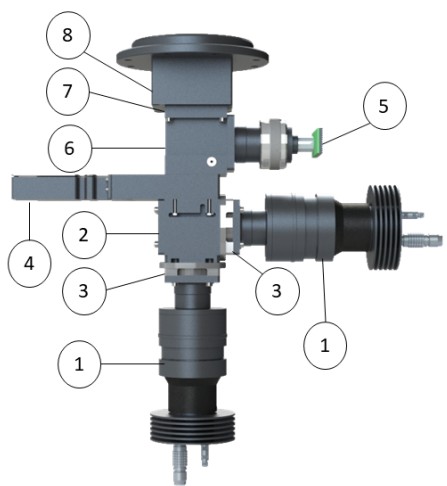

**Figure 4.** The detection unit after telescope A, containing the optical elements that are used for the detection of the polarization state of the backscattered light, with the signals recorded at the two APDs (1). It also contains a $PBS$ (2), followed by cleaning polarizing sheet filters (3), a shutter for dark measurements (4), a camera for the alignment (5), a turning mirror for redirecting the light to the camera (6), an interference filter for the reduction of the background light (7), and a mechanical rotator (8) for accurately rotating a $HWP$ for the system calibration (Section 4). The detection unit after telescope B is the same, with a $QWP$ placed before the $PBS$.

     The lidar head is protected from rain and dust, with covers and with special glass windows in front of the lasers and
the telescopes (Fig. 3, left). The covers can be easily removed to allow access to the internal parts of the head. Moreover, thermoelectric coolers are installed inside the head in order to stabilize the internal temperature, and to provide tolerance to ambient temperatures up to 45 °C.

## 2.2   The positioner

     The positioner enables the lidar head to move along different zenith and azimuth angles. The positioning of the head at various
viewing angles is controlled manually. Due to constrains from the umbilical tubes, the head can be moved along $-10^o$ to $+90^o$ from the zenith, and at $-150^o$ to $+150^o$ around the vertical. The positioner consists of two side arms and a base (Fig. 5) which

can be manually rotated. For changing the zenith angle, the one of the side arms is driving and the other is free. A break on the free arm reduces the backlash.

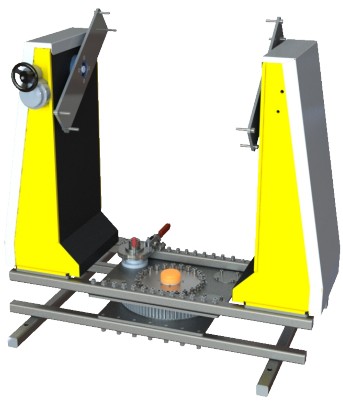

**Figure 5.** The fork-type positioner of the lidar head.

## 2.3 The electronics compartment

The electronics compartment (Fig. 6) contains the power supplies of the two lasers, the LPC, the LICEL rack containing the TRs for digitizing and recording the signals from the APDs, and the Master Trigger Control Unit that synchronizes the emission of the two lasers and the acquisition of the backscattered signals. Moreover, it contains a UPS and a precipitation sensor. The UPS can provide power to the system for about one hour, in case of power failure. This is enough time for a proper cool down of the lasers and shutting down of the system. The precipitation sensor causes shutdown of the lidar when precipitation is 140 detected.

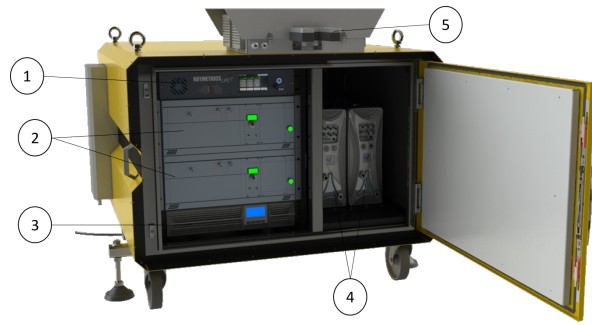

**Figure 6.** The enclosure with 1) the LPC unit, 2) the LICEL rack with the TRs and the Master Trigger Control, 3) the UPS, 4) the power supplies of the lasers, and 5) the precipitation sensor.

The synchronization of this complicated lidar system with two lasers emitting interleavingly and with their backscattered signals recorded interleavingly, requires a sophisticated triggering system. We use a master trigger control unit, produced by Licel GmbH (Fig. 7) that utilizes two trigger generators for the synchronization of the emission of the lasers and of the acquisition of the signals. As shown in Fig. 7b, the first trigger generator produces a pulse that starts the flash lamp. The laser builds up its maximum energy for 160 $\mu$sec, and then a second pulse turns on the active Q-switch, which allows the release of the laser beam pulse. In the meantime, a third pulse triggers the acquisition of the backscattered light from laser A. Pre-trigger measurements are acquired until the emission of the laser A beam pulse. The same sequence is performed from the second trigger generator for laser B, starting 10 ms later, in order to avoid the recording of overlapping photons from the backscattered light of laser A. The duty cycle of lasers A and B is $\sim$ 12 ms and the time between each cycle is 50 ms (fixed due to the repetition rate of the lasers of 20 Hz), with 38 ms idle time.

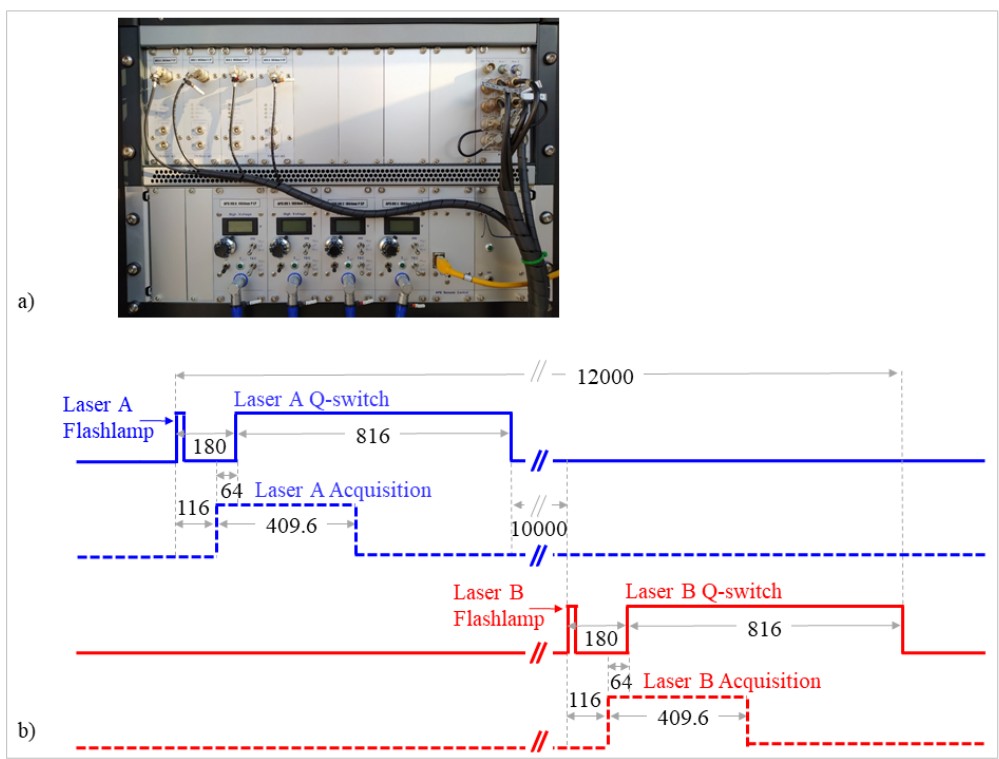

**Figure 7.** a) The LICEL rack with the TRs and the Master Trigger Control Unit. b) The pulses from the Master Trigger Control Unit that synchronize the emission of the two lasers and the acquisition of the backscattered signals. The lengths of the pulses are in μs.

The lidar system is controlled from the LPC unit. This is an "enhanced" embedded computer with specific I/Os that fits the lidar requirements, providing several ethernet interfaces that make the controlling (local or remote) of the lidar easy and safe. The LPC controls all lidar sub-components (e.g. the lasers, data acquisition systems), along with any auxiliary equipment used by the lidar system (e.g., the precipitation sensor, temperature and humidity sensors, cameras for the alignment). Additionally, it controls the mechanical rotators of the optical elements used for calibration purposes (Section 4), and it stores the acquired raw measurements. The precipitation sensor (Fig. 6) provides information about precipitation conditions and causes shutdown of the lidar when precipitation is detected. Moreover, several external easy accessible push buttons are connected to the LPC and can be used by the operators to shut down the lasers in case of emergency.

## 3  Emission and detection design based on the measurement strategy

The core of the new lidar system is its emission and detection design, based on our measurement strategy for monitoring the oriented dust in the atmosphere. Our approach is different from the measurement strategy of previous works, which either focus

on the retrieval of the individual elements of the backscatter matrix of the oriented particles utilizing moving elements in the system (e.g., Kaul et al., 2004), or use complicated designs that are difficult to be calibrated effectively (e.g., Geier and Arienti, 2014). We choose to avoid both, in order to achieve robust measurements along with their effective calibration. Moreover, most

of the previous works utilize visible light measurements whereas we use near infrared light measurements at $1064\,\mathrm{nm}$, to better probe the larger dust particles (Gasteiger and Freudenthaler, 2014; Burton et al., 2016).

Figure 2 shows the simplest design of a "two lasers/two telescopes/four detectors" lidar system that is able to detect the elliptically-polarized backscattered light from oriented particles in the atmosphere without using any moving parts for the emission or the detection of light. The linear polarization of the backscattered signal is detected using a linear-polarization-

analyzer (a $PBS$) in the detection unit after telescope A, and the circular polarization of the backscattered signal is detected using a circular-polarization-analyzer (a $QWP$ followed by a $PBS$) in the detection unit after telescope B. The calibration methodology is based on the solutions introduced by Freudenthaler (2016) for EARLINET lidar systems, as well as on new methodologies tailored for the detection of oriented particles, presented in Section 4.

Instead of retrieving the individual off-diagonal elements of the backscatter matrix, we aim for measurements that are

combinations of only the off-diagonal elements of the backscatter matrix that will be nonzero only in case of oriented particles. This way we acquire direct measurements of dust orientation, in the form of flags of "yes" or "no" particle orientation. This first-level information of the oriented dust in the atmosphere is straightforward to achieve, since it does not require any inversion procedure. Moreover, it is important to have, considering that the possibility of dust orientation has not been extensively investigated in the Earth's atmosphere, even at this elementary level. To achieve this, laser A should emit linearly-polarized

light at $45^{o}$, as discussed in detail below.

Along with the measurements of "orientation flags", we use the measurements from laser B in order to acquire additional information for the particle orientation properties, as the particle orientation angle and the percentage of oriented particles in the atmosphere, as well as information on dust microphysics, i.e. an estimation of the particle size and refractive index. These are parameters that are necessary to have in order to explain the phenomenon of dust orientation in more detail. The

methodology for defining the optimum measurements includes extensive simulations for different atmospheric scenarios and machine learning tools. Briefly, the backscattered light is simulated for different mixtures of dust particles with realistic sizes and irregular shapes, including cases with random and preferential particle orientation. We investigate a large number of possible polarizations for laser B, and we evaluate their information content based on the performance of the corresponding neural network retrievals that use the simulated lidar measurements to retrieve the oriented dust microphysical properties. This

is an ongoing work, with the first results identifying that the emission from laser B should be elliptically-polarized, with the angle of the polarization ellipse at $5.6^{o}$ and degree of linear polarization of $0.866$.

This Section provides the methodology we followed to define the properties of the optical elements in the emission and the detection units of the lidar system, so as to fulfil the technical and scientific objectives of our measurement strategy. Considering the layout in Fig. 2 the only "free" parameters that we need to define are the polarization state of the light from the emission

units of laser A and B, and the position of the $QWP_{TB}$ in the detection unit after telescope B (the $HWP_{TA}$ and $HWP_{TB}$ are used for calibration purposes (see Section 4 and Section S4 in the Supplement)).

The measured signal $I^*_{i\_k\_s}$ for laser $i = LA$, $LB$, at the detection unit after telescope $k = TA$, $TB$, at the detector $s = T$, $R$ ("transmitted" and "reflected" channel after the $PBS_k$, respectively), is shown in Eq. 1.

$$I^*_{i\_k\_s} = \eta_{s\_k} \boldsymbol{M}_{i\_k}[(\mathbf{F} + \mathbf{G})\boldsymbol{i}_i + \boldsymbol{i}_g] + y_{N_{i\_k\_s}} \tag{1}$$

$$\boldsymbol{M}_{i\_TA} = E_{i\_TA}\boldsymbol{e}\mathbf{M}_{s\_TA}\mathbf{M}_{O\_TA} \tag{2}$$

$$\boldsymbol{M}_{i\_TB} = E_{i\_TB}\boldsymbol{e}\mathbf{M}_{s\_TB}\mathbf{M}_{QW\_TB}\mathbf{M}_{O\_TB} \tag{3}$$

In Eq. 1, $\eta_{s\_k}$ is the amplification of the signals of lasers at $s = T$ or $R$ detector of the detection unit after telescope $k$, $\boldsymbol{M}_{i\_k}$ is a row vector expressing the measured polarization at the detection unit after telescope $k$, $\mathbf{F}$ and $\mathbf{G}$ are the backscatter Stokes phase matrices of the dust particles and of the gas molecules, respectively, at a certain range in the atmosphere, and $\boldsymbol{i}_i$
and $\boldsymbol{i}_g$ are the Stokes vector of the light from the emission unit of laser $i$, and the Stokes vector of the background skylight, respectively. Equations 2 and 3 describe in more detail the vectors $\boldsymbol{M}_{i\_k}$: $E_{i\_k} = A_k O_{i\_k} T(0,r)^{-2} E_{oi}$, where $A_k$ is the area of the telescope $k$, $O_{i\_k}$ is the overlap function of the laser beam receiver field-of-view with range 0-1 (for laser $i$ and telescope $k$), $T(0,r)$ is the transmission of the atmosphere between the lidar at range $r = 0$ and a specific range in the atmosphere, and $E_{oi}$ is the pulse energy of laser $i$. $T(0,r)$ is simplified to a scalar, since the polarization effect due to the transmission (i.e.
dichroism) is deemed to be small (Ulanowski et al., 2007). $\boldsymbol{e} = [1,0,0,0]$ denotes the measurement of only the intensity of light reaching the APDs. $\mathbf{M}_{s\_k}$ is the Mueller matrix of the $PBS_k$ followed by cleaning polarizing sheet filters, $\mathbf{M}_{O\_k}$ is the Mueller matrix of the receiver optics (i.e. telescope $k$, collimating lenses, bandpass filter), and $\mathbf{M}_{QW\_TB}$ is the Mueller matrix of the $QWP_{TB}$. For the explicit definition of the Muller matrices in Eq. 1, 2 and 3, see Section S1 in the Supplement. $y_{N_{i\_k\_s}}$ is the electronic background of the $s = T$ or $R$ detector at the detection unit after telescope $k$, for detecting the backscattered
signal of laser $i$.

We simplify Eq. 1 as shown in Eq. 4, considering measurements that are free from the sunlight and electronic background. We use vector $\boldsymbol{p}_{i\_k}$ with elements calculated by the elements of $\boldsymbol{i}_i$ (emitted polarization) and $\boldsymbol{M}_{i\_k}$ (detected polarization), as shown in Eq. 5. $\boldsymbol{f}$ and $\boldsymbol{g}$ are the vectorized scattering matrices $\mathbf{F}$ and $\mathbf{G}$, respectively ($f_{m+4(n-1)} = F_{mn}$ and $g_{m+4(n-1)} = G_{mn}$).

$$I_{i\_k\_s} = \eta_{s\_k}\boldsymbol{p}_{i\_k}(\boldsymbol{f} + \boldsymbol{g}) \tag{4}$$

$$p_{i\_k_{m+4(n-1)}} = M_{i\_k_m} i_{i_n} \tag{5}$$

Performing the Mueller matrix calculations, $I_{i\_TA\_s}$ and $I_{i\_TB\_s}$ can be written as a function of the Stokes parameters of the light from the emission units of the lasers $\boldsymbol{i}_i = [I_i, Q_i, U_i, V_i]^T$, the fast-axis-angle $\phi_{TB}$ of the $QWP_{TB}$ with the reference plane, and the backscatter Stokes phase matrix elements, as shown in Eq. A1 and A2 in Appendix A. As can be deduced by these equations, in order to use laser A to achieve measurements that contain only the off-diagonal elements of the backscatter matrix, the following conditions must be met: $Q_{LA} = 0$ for $I_{LA\_TA\_s}$, $V_{LA} = 0$ for $I_{LA\_TB\_s}$, and $\phi_{TB} = 45^o$. Thus, the Stokes vector of the light from the emission unit of laser A should be $45^o$-linearly polarized, with Stokes vector $\boldsymbol{i}_{LA} = [1, 0, 1, 0]^T$. This is achieved using the $HWP_{LA}$ in front of laser A as discussed in Section 4.2. Moreover, the fast-axis-angle of the $QWP_{TB}$ should be at $\phi_{TB} = 45^o$.

For calibration reasons, we use the ratios of the measurements of the reflected and transmitted channels after the $PBS_k$ (Eq. 6 and 7). Then, the calibrated backscatter signal ratios of laser A provide direct flags of particle orientation ($F_{LA\_TA}$ and $F_{LA\_TB}$ in Eq. 9 and 10, respectively), when their values are $\neq 1$.

$$\frac{I_{LA\_TA\_R}}{I_{LA\_TA\_T}} = \eta_{TA} \frac{1 - f_{12} + f_{13} - f_{23} + g_{11}}{1 + f_{12} + f_{13} + f_{23} + g_{11}} \tag{6}$$

$$\frac{I_{LA\_TB\_R}}{I_{LA\_TB\_T}} = \eta_{TB} \frac{1 + f_{13} + f_{14} - f_{34} + g_{11}}{1 + f_{13} - f_{14} + f_{34} + g_{11}} \tag{7}$$

The calibration factors $\eta_{TA}$ and $\eta_{TB}$ are derived as shown in Section 5. Due to the $HWP_{TB}$ in the detection unit after telescope B (used for checking for systematic errors in measurements, as discussed in Section S4 in the Supplement), Eq. 7 changes to Eq. 8.

$$\frac{I_{LA\_TB\_R}}{I_{LA\_TB\_T}} = \eta_{TB} \frac{1 + f_{13} - f_{14} + f_{34} + g_{11}}{1 + f_{13} + f_{14} - f_{34} + g_{11}} \tag{8}$$

The orientation flags $F_{LA\_TA}$ and $F_{LA\_TB}$ are then provided from Eq. 9 and 10.

$$F_{LA\_TA} = \frac{1}{\eta_{TA}} \frac{I_{LA\_TA\_R}}{I_{LA\_TA\_T}} \tag{9}$$

$$F_{LA\_TB} = \frac{1}{\eta_{TB}} \frac{I_{LA\_TB\_R}}{I_{LA\_TB\_T}} \tag{10}$$

Having achieved signals that provide direct orientation flags with laser A, we use laser B to increase the information content of the measurements in terms of dust orientation properties (e.g. angle and percentage of oriented particles in the atmosphere)

and of dust microphysical properties (e.g. size and refractive index). As discussed above, the derivation of the optimum polarization state of $i_{LB}$ is still ongoing, with the elliptical polarization provided herein ($i_{LB} = [1, 0.85, 0.17, 0.5]^T$) to be a first estimation. The elliptical polarization of $i_{LB}$ is set with the $QWP_{LB}$ and the $HWP_{LB}$ in front of laser B (Fig. 2), as discussed in Section 4.3. The corresponding signal ratios are shown in Eq. A11 and A12 in Appendix A.

## 4   Definition of the polarization of the emitted and detected light with respect to the horizon

The polarization of the light emitted and detected by the system should be defined with respect to the horizon, so as the retrieved properties of the oriented particles are defined with respect to the horizon. This is done by first leveling the head of the lidar along the horizon using a spirit level, which then enables us to use the frame of the lidar head as the reference coordinate system. The "frame coordinate system" (Fig. 8a) is a right-handed coordinate system, with $x_F$-axis parallel to the horizon and $z_F$-axis pointing in the propagation direction of the emitted light from lasers A and B, considering that both lasers are parallel.

The optical elements are considered to be well aligned with eachother in the detection units after telescopes A and B (because their holders are manufactured and assembled in a mechanical workshop with high accuracy), but the detection units are possibly rotated around the optical axis with respect to the frame coordinate system by angles $\omega_{TA}$ and $\omega_{TB}$, respectively (Fig. 8b and c). The Stokes vectors of the light collected at telescopes A and B are consequently described including a multiplication with the rotation matrices $\mathbf{R_{TA}}(-\omega_{TA})$ and $\mathbf{R_{TB}}(-\omega_{TB})$, respectively (see Eq. S.5.1.7 in Freudenthaler (2016)), which affects the measurements of the polarized components after $PBS_{TA}$ (Eq. A3 and A5), but not after $PBS_{TB}$ (Eq. A7 and A8). The rotation of the detection unit after telescope A is corrected using the $HWP_{TA}$, as shown in Section 4.1.

The "$DU_{TA}$ coordinate system" and the "$DU_{TB}$ coordinate system" in Fig. 8b and c are the right-handed coordinate systems of the detection units after telescopes A and B, respectively. The $x_{DU_{TA}}$ and $y_{DU_{TA}}$ axis coincide with the incidence plane of $PBS_{TA}$, and the $x_{DU_{TB}}$ and $y_{DU_{TB}}$ axis coincide with the incidence plane of $PBS_{TB}$.

In Appendix A, equations A3-A8 show the formulation of Eq. A1 and A2 for $I_{i\_k\_s}$ with respect to the frame coordinate system, taking into account all the optical elements of the system, along with the rotation of the detection units after telescopes A and B. The analytic derivations of Eq. A3-A8 are provided in Section S2 of the Supplement.

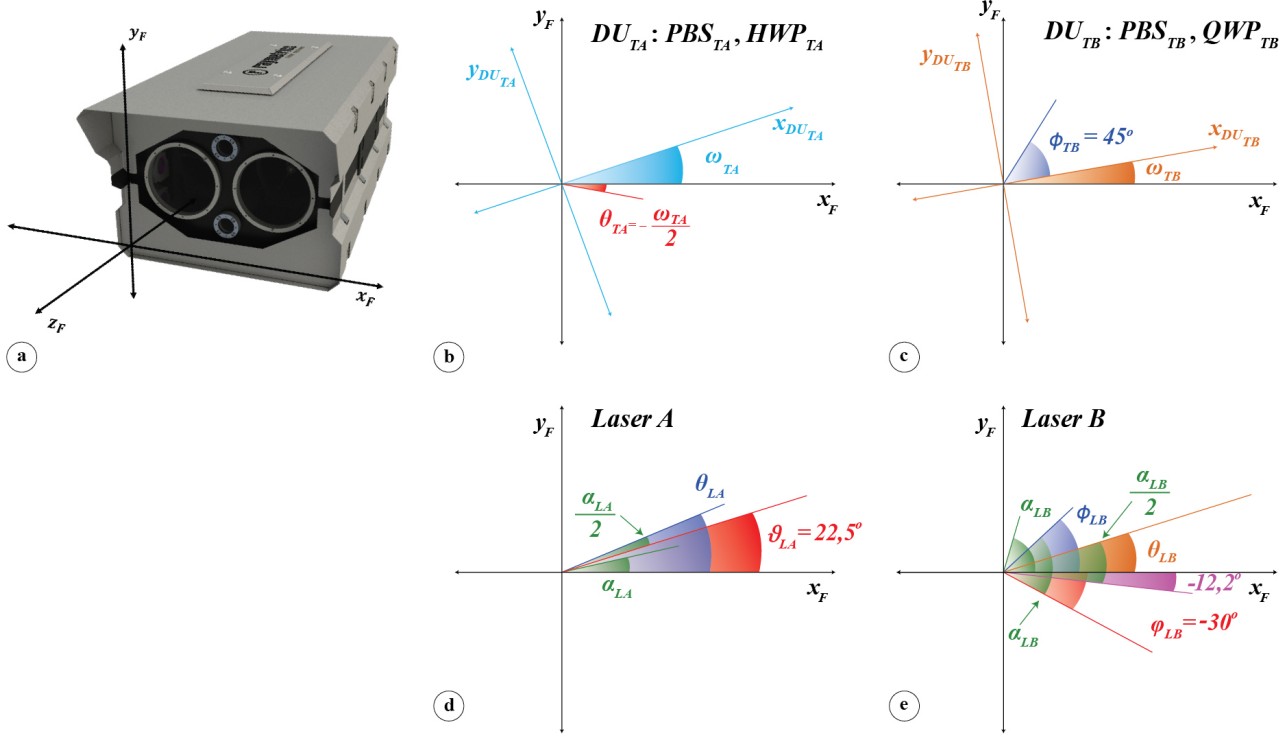

**Figure 8.** a) The "frame coordinate system" (black) is the reference coordinate system with $x_F$-axis parallel to the horizon. b) The "$DU_{TA}$ coordinate system" (light blue) is the coordinate system of the detection unit after telescope A, which is rotated with respect to the frame coordinate system by an angle $\omega_{TA}$. The effect of this rotation on the signals is corrected using $HWP_{TA}$, placed at $\theta_{TA} = -\frac{\omega_{TA}}{2}$ (red) with respect to the $x_F$-axis. c) The "$DU_{TB}$ coordinate system" (orange) is the coordinate system of the detection unit after telescope B, which is rotated with respect to the frame coordinate system by an angle $\omega_{TB}$ . The rotation does not affect the measured signals. The $QWP_{TB}$ before $PBS_{TB}$, is placed at $\phi_{TB} = 45^o$ with respect to the $x_{DU_{TB}}$-axis. d) The light emitted directly from laser A is linearly-polarized with unknown angle of polarization $\alpha_{LA}$. As shown in Eq. 11, using the $HWP_{LA}$ with fast-axis-angle $\theta_{LA} = 22.5^o + \frac{\alpha_{LA}}{2}$, we produce the light emitted from the emission unit of laser A with angle of polarization $2\vartheta_{LA} = 45^o$. e) The light emitted directly from laser B is linearly-polarized with unknown angle of polarization $\alpha_{LB}$. As shown in Eq. 12, using the $QWP_{LB}$ with with fast-axis-angle $\phi_{LB} = \alpha_{LB} - 30^o$, and the $HWP_{LB}$ with fast-axis-angle $\theta_{LB} = \frac{\alpha_{LB}}{2} - 12.2^o$, we produce the elliptically-polarized light emitted from the emission unit of laser B with angle of polarization $5.6^o$ and degree of linear polarization $0.866$.

The Stokes vector of the light from the emission unit of laser A and B is provided by $\boldsymbol{i}_{LA}$ (Eq. 11) and $\boldsymbol{i}_{LB}$ (Eq. 12), respectively. The light emitted directly from laser A ($\boldsymbol{i}_{lsr\_LA}$) and laser B ($\boldsymbol{i}_{lsr\_LB}$) is considered to be $100\%$ linearly-polarized, with angle of polarization ellipse with respect to the frame coordinate system $\alpha_{LA}$ and $\alpha_{LB}$, respectively, i.e. $\boldsymbol{i}_{lsr\_LA}(\alpha_{LA}) =$
$\begin{bmatrix} 1 & c_{2\alpha_{LA}} & s_{2\alpha_{LA}} & 0 \end{bmatrix}^T$ in Eq. 11, and $\boldsymbol{i}_{lsr\_LB}(\alpha_{LB}) = \begin{bmatrix} 1 & c_{2\alpha_{LB}} & s_{2\alpha_{LB}} & 0 \end{bmatrix}^T$ in Eq. 12. The angles $\alpha_{LA}$ and $\alpha_{LB}$ are

unknown a-priori. The polarization of the light from the whole emission unit is defined according to the position of the optical elements in front of the lasers with respect to the frame coordinate system, i.e. the fast-axis-angle $\theta_{LA}$ of the $HWP_{LA}$ in front of laser A, and the fast-axis-angle $\phi_{LB}$ of $QWP_{LB}$ followed by the $HWP_{LB}$ with fast-axis-angle $\theta_{LB}$ in front of laser B (Fig. 8d and e; Eq. 11 and 12).

In order to simplify Eq. 11 and 12 we use the angles $\vartheta_{LA} = \theta_{LA} - \frac{\alpha_{LA}}{2}$ and $\varphi_{LB} = \phi_{LB} - \alpha_{LB}$. From Eq. 11 and 12 we deduce: $\vartheta_{LA} = 22.5^o$ and $\theta_{LA} = 22.5^o - \frac{\alpha_{LA}}{2}$ (Fig. 8d), $\varphi_{LB} = -30^o$, $\phi_{LB} = \alpha_{LB} - 30^o$ and $\theta_{LB} = \frac{\alpha_{LB}}{2} - 12.2^o$ (Fig. 8e).

$$i_{LA} = \mathbf{M}_{HW\_LA}(\theta_{LA})i_{lsr\_LA}(\alpha_{LA}) = \begin{bmatrix} 1 \\ c_{(4\theta_{LA}-2\alpha_{LA})} \\ s_{(4\theta_{LA}-2\alpha_{LA})} \\ 0 \end{bmatrix} = \begin{bmatrix} 1 \\ c_{4\vartheta_{LA}} \\ s_{4\vartheta_{LA}} \\ 0 \end{bmatrix} = \begin{bmatrix} 1 \\ 0 \\ 1 \\ 0 \end{bmatrix} \tag{11}$$

$$i_{LB} = \mathbf{M}_{HW\_LB}(\theta_{LB})\mathbf{M}_{QW\_LB}(\phi_{LB})i_{lsr\_LB}(\alpha_{LB}) = \begin{bmatrix} 1 \\ c_{2(\phi_{LB}-\alpha_{LB})}c_{(4\theta_{LB}-2\phi_{LB})} \\ c_{2(\phi_{LB}-\alpha_{LB})}s_{(4\theta_{LB}-2\phi_{LB})} \\ -s_{2(\phi_{LB}-\alpha_{LB})} \end{bmatrix} = \begin{bmatrix} 1 \\ c_{2\varphi_{LB}}c_{(4\theta_{LB}-2\phi_{LB})} \\ c_{2\varphi_{LB}}s_{(4\theta_{LB}-2\phi_{LB})} \\ -s_{2\varphi_{LB}} \end{bmatrix} = \begin{bmatrix} 1 \\ 0.85 \\ 0.17 \\ 0.5 \end{bmatrix} \tag{12}$$

## 4.1   Correction of the signal I$_{i\_TA\_s}$, due to the rotation of the detection unit after telescope A

The analytic formulas provided for the signals $I_{i\_TA\_s}$ in Appendix A show that the rotation of the detection unit after telescope A changes the signals $I_{LA\_TA\_s}$ (Eq. A3) and $I_{LB\_TA\_s}$ (Eq. A5). In order to correct for this effect, we have to set the fast-axis-angle of the $HWP_{TA}$ at $\theta_{TA} = -\frac{\omega_{TA}}{2}$ with respect to the $x_{DU_{TA}}$-axis (Fig. 8b), so that $c_{(4\theta_{TA}+2\omega_{TA})} = 1$ and $s_{(4\theta_{TA}+2\omega_{TA})} = 0$ in Eq. A3 and A5. After this correction, $I_{LA\_TA\_s}$ and $I_{LB\_TA\_s}$ are provided by Eq. A4 and Eq. A6, respectively.

Since the value of $\theta_{TA}$ with respect to the $x_{DU_{TA}}$-axis is unknown a priori, we assume that it deviates from the desired value by an unknown angle $\varepsilon_{TA}$, thus $\theta_{TA} = \frac{-\omega_{TA}}{2} + \varepsilon_{TA}$. We derive $\varepsilon_{TA}$ by using the measurements from laser A, after placing a linear polarizer in front of the window of laser A at $45^o$ from $x_F$-axis (Fig. 9), and rotating the $HWP_{TA}$ in order to perform a methodology similar to the "$\Delta 90^o$ calibration" of Freudenthaler (2016), as shown in Fig. 10 and described in detail in Appendix B. The methodology is applicable only when there are only randomly-oriented particles in the atmosphere.

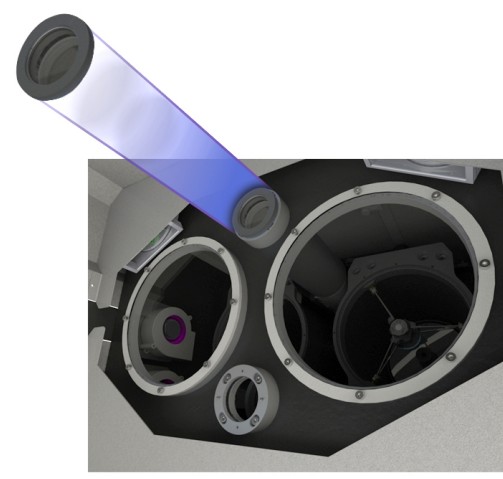

**Figure 9.** Linear polarizer in front of the window of laser A, placed at $45^o$ from $x_F$-axis.

Use $HWP_{TA}$ to correct the rotation of the detection unit after telescope A, with respect to the horizon

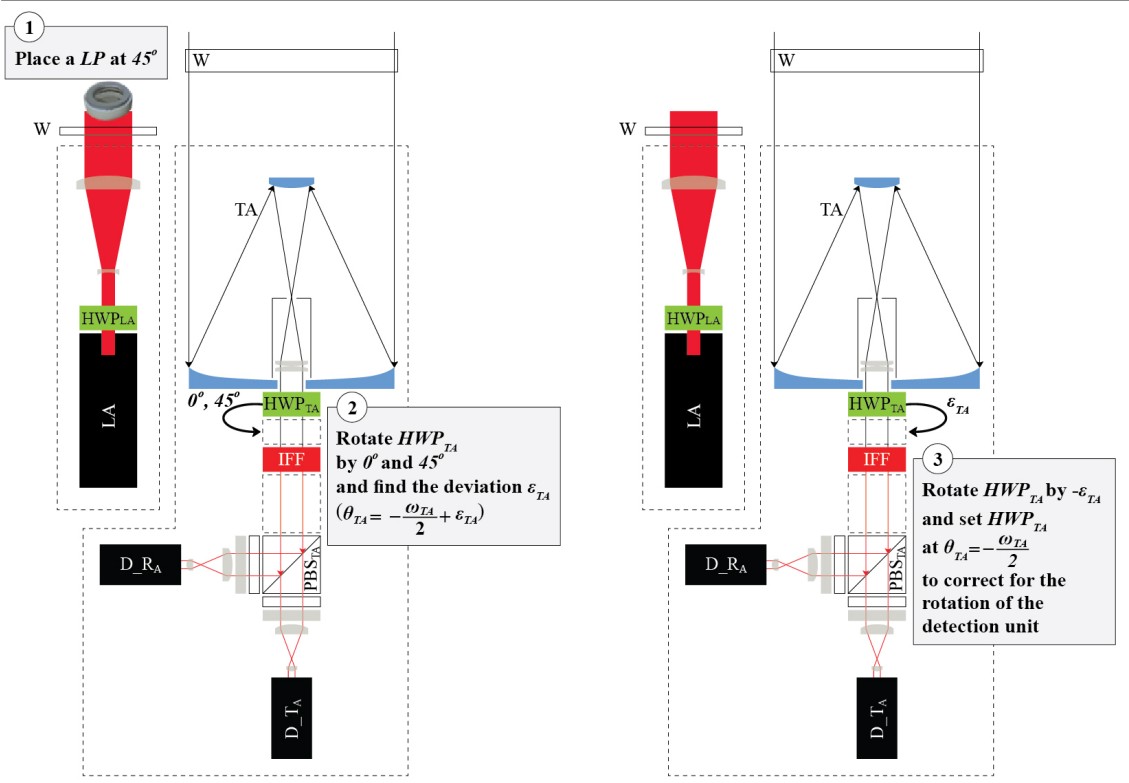

**Figure 10.** Methodology for correcting the measurements $I_{i\_TA\_s}$ due to the rotation of the detection unit after telescope A.

**4.2 Definition of the polarization of the light from the emission unit of laser A with respect to the horizon**

In order to set the linear polarization of the light from the emission unit of laser A at $45^o$ degrees with respect to the horizon, as discussed in Section 3, we have to set $\vartheta_{LA} = 22.5^o$ with respect to $x_F$-axis (Eq. 11; Fig. 8d). Since the value of $\vartheta_{LA}$ is unknown a priori, we assume that it is equal to an unknown angle $\varepsilon_{LA}$. We derive $\varepsilon_{LA}$ by performing the "$\Delta 90^o$ calibration" of Freudenthaler (2016) by rotating the $HWP_{LA}$ in front of laser A, as shown in Fig. 11 and discussed in detail in Appendix 295 C. Then, we rotate the $HWP_{LA}$ by $22.5^o - \varepsilon_{LA}$ and set $\vartheta_{LA} = 22.5^o$. ($\vartheta_{LA} = \theta_{LA} - \frac{\alpha_{LA}}{2}$ (Eq. 11), thus in order to change $\vartheta_{LA}$ by angle $x$, it is sufficient to rotate the $HWP_{LA}$ and change its fast-axis angle $\theta_{LA}$ by angle $x$.)

Our methodology considers that the atmosphere consists of only randomly-oriented particles and that we have corrected the effect of the rotation of the detection unit after telescope A on signals $I_{i\_TA\_s}$ (Section 4.1).

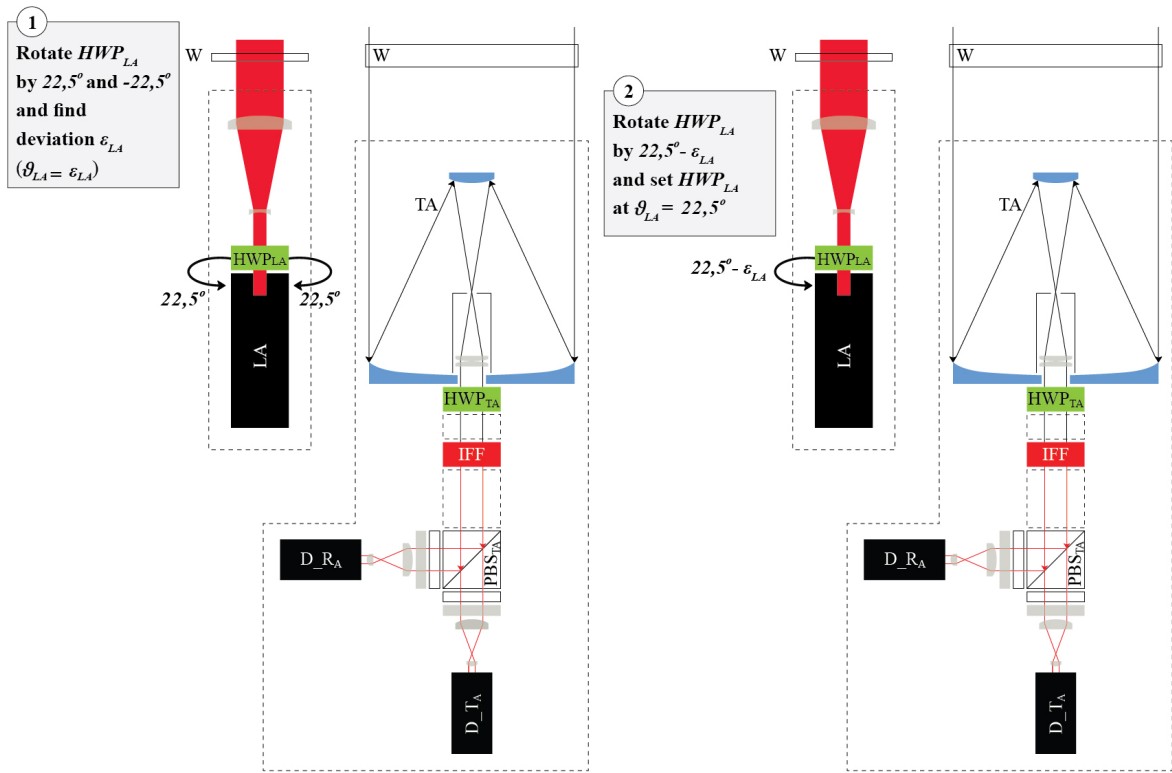

**Figure 11.** Methodology for defining the polarization of the light from the emission unit of laser A, with respect to the horizon.

### 4.3 Definition of the polarization of the light from the emission unit of laser B with respect to the horizon

The desired elliptical polarization of the light from the emission unit of laser B with respect to the horizon is shown in Eq. 13, with angle of the polarization ellipse $\alpha_{em} = 5.6^o$ and degree of linear polarization $b_{em} = 0.866$. $b_{em} = c_{2\varphi_{LB}}$, thus in order to set it to the desired value, we rotate the $QWP_{LB}$ and change $\varphi_{LB}$ appropriately ($\varphi_{LB} = \phi_{LB} - \alpha_{LB}$ (Eq. 12), thus for changing $\varphi_{LB}$ by angle $x$, it is sufficient to rotate the $QWP_{LB}$ and change its fast-axis angle $\phi_{LB}$ by angle $x$.). After we have set $b_{em}$, we set $\alpha_{em} = 2\theta_{LB} - \phi_{LB}$ to the desired value, by rotating the $HWP_{LB}$ in front of laser B. The angles $\alpha_{LB}$, $\varphi_{LB}$,
$\phi_{LB}$ and $\theta_{LB}$ are shown in Fig. 8e.

Our methodology is described in Fig. 12. We consider randomly-oriented particles in the atmosphere and we use the measurements of laser B at the detection unit after telescope A. The detection unit is aligned with the frame coordinate system (Section 4.1). First, we consider that the polarization of the light from the emission unit of laser B with respect to the frame coordinate system is unknown, with unknown angle of the polarization ellipse $\alpha$ and degree of linear polarization $b$. In order
to set them to $\alpha_{em}, b_{em}$, we perform the following steps:

1. We derive $\alpha$ and $b$ by turning the $HWP_{LB}$ by appropriate angles, using a similar methodology to the "$\Delta 90^o$ calibration" of Freudenthaler (2016), as shown in step 1 and 2 in Fig. 12, and in detail in Appendix D.

2. We change $\varphi_{LB} = \frac{acos(b)}{2}$ by turning the $QWP_{LB}$ by $\frac{acos(b_{em})-acos(b)}{2}$. Then, $\varphi_{LB\_new} = \frac{acos(b_{em})}{2}$ and the degree of linear polarization is set to the desired value $b_{em}$ (step 3 in Fig. 12).

3. The turning of $QWP_{LB}$ in (2), changes the angle of the polarization ellipse to an unknown value of $\alpha_{new}$. We calculate $\alpha_{new}$ using the methodology in (1) (step 4 in Fig. 12).

4. After deriving $\alpha_{new}$, we set the angle of the polarization ellipse to the desired value $\alpha_{em}$, by turning $HWP_{LB}$ by $\frac{\alpha_{em}-\alpha_{new}}{2}$ (step 5 in Fig. 12).

$$
\boldsymbol{i}_{LB} = \begin{bmatrix} 1 \\ c_{2\varphi_{LB}} c_{(4\theta_{LB}-2\phi_{LB})} \\ c_{2\varphi_{LB}} s_{(4\theta_{LB}-2\phi_{LB})} \\ -s_{2\varphi_{LB}} \end{bmatrix} = \begin{bmatrix} 1 \\ b_{em} c_{2\alpha_{em}} \\ b_{em} s_{2\alpha_{em}} \\ \sqrt{1-b_{em}^2} \end{bmatrix} = \begin{bmatrix} 1 \\ 0.85 \\ 0.17 \\ 0.5 \end{bmatrix}
\tag{13}
$$

Use $HWP_{LB}$ and $QWP_{LB}$ to define the polarization of laser B with respect to the horizon

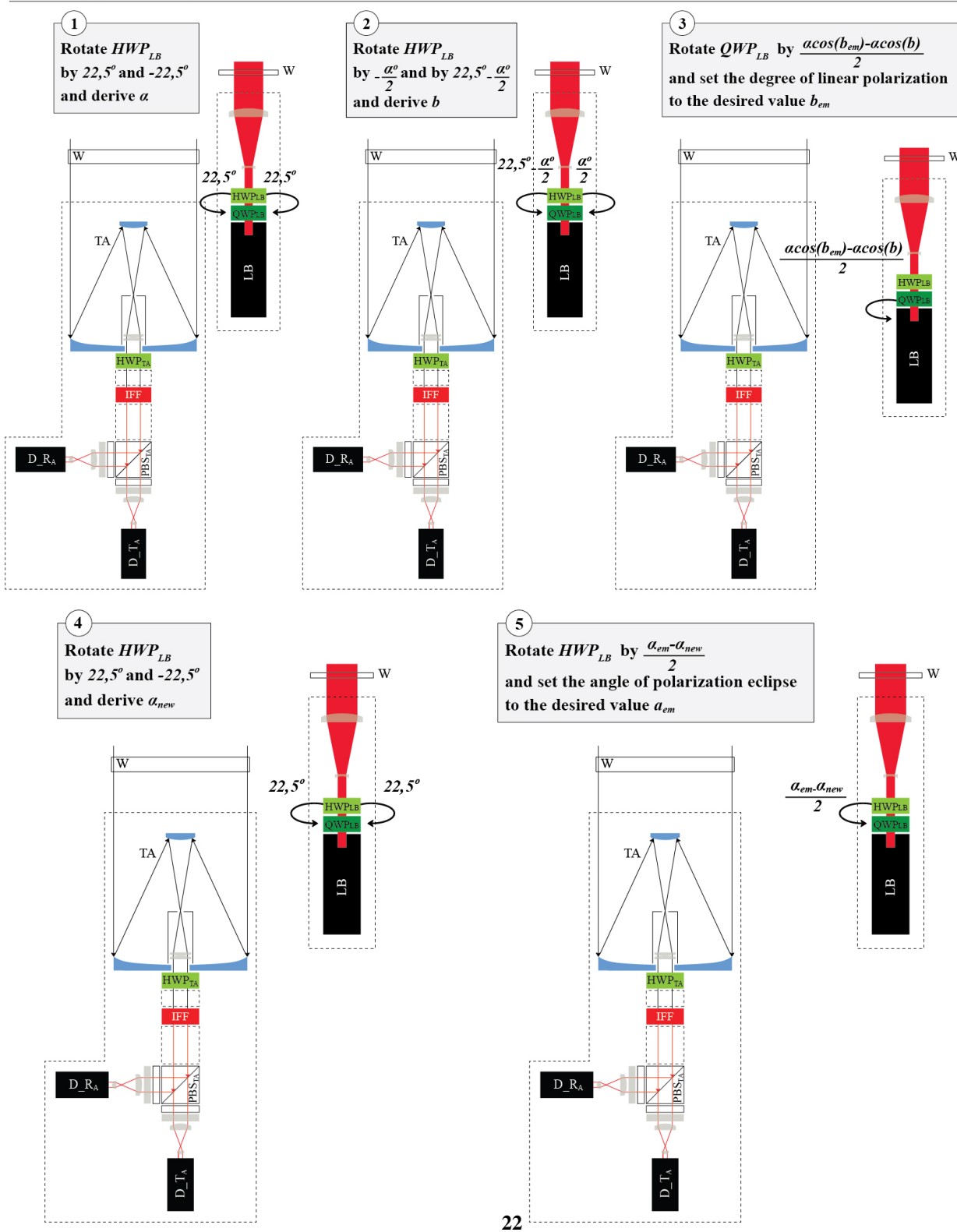

**Figure 12.** Methodology for defining the polarization of the emitted light from laser B with respect to the horizon.

## 5 The derivation of the calibration factors $\eta_{TA}$ and $\eta_{TB}$

The calibration factors $\eta_{TA}$ and $\eta_{TB}$ are derived considering randomly-oriented particles in the atmosphere.

We calculate $\eta_{TA}$ using the ratio of the signals from laser A at the detection unit after telescope A, considering that the effect of its rotation on the signals is corrected (Section 4.1). $\eta_{TA} = \frac{\eta_{R\_TA} T_{R\_TA}}{\eta_{T\_TA} T_{T\_TA}}$ is calculated as shown in Eq. 14, using Eq. A4 in Appendix A, with zero off-diagonal elements.

$$\frac{I_{LA\_TA\_R}}{I_{LA\_TA\_T}} = \frac{\eta_{R\_TA} T_{R\_TA} \left[ f_{11} + g_{11} \right]}{\eta_{T\_TA} T_{T\_TA} \left[ f_{11} + g_{11} \right]} \Rightarrow \eta_{TA} = \frac{I_{LA\_TA\_R}}{I_{LA\_TA\_T}} \tag{14}$$

We derive the calibration factor $\eta_{TB} = \frac{\eta_{R\_TB} T_{R\_TB}}{\eta_{T\_TB} T_{T\_TB}}$ using the ratio of the signals from laser A at the detection unit after telescope B, as shown in Eq. 15, using Eq. A7 in Appendix A, with zero off-diagonal elements.

$$\frac{I_{LA\_TB\_R}}{I_{LA\_TB\_T}} = \frac{\eta_{R\_TB} T_{R\_TB} \left[ f_{11} + g_{11} \right]}{\eta_{T\_TB} T_{T\_TB} \left[ f_{11} + g_{11} \right]} \Rightarrow \eta_{TB} = \frac{I_{LA\_TB\_R}}{I_{LA\_TB\_T}} \tag{15}$$

## 6 The derivation of the volume linear depolarization ratio (VLDR)

The VLDR ($\delta$) is a useful optical parameter for comparing the measurements of the new polarization lidar with measurements from the commonly-used polarization lidars, which emit linearly-polarized light and measure the corresponding cross- and parallel-polarized components of the backscattered light. $\delta$ is calculated using the atmospheric polarization parameter $a$ as shown in Eq. 16.

$$\delta = \frac{1 - a}{1 + a} \tag{16}$$

We derive $\delta$ considering an atmosphere containing only randomly-oriented particles. Moreover, we consider that the effect of the rotation of the detection unit after telescope A with respect to the frame coordinate system is corrected (Section 4.1) and that the calibration factor $\eta_{TA}$ is calculated using measurements of laser A, as shown in Section 5.

We turn $HWP_{TA}$ by $22.5^o$, so as $\theta_{TA} = -\frac{\omega_{TA}}{2} + 22.5^o$. Then, using $\theta_{TA}$ in Eq. A3 and $a = \frac{f_{22} + g_{22}}{f_{11} + g_{11}}$, we derive $I_{LA\_TA\_T}$ and $I_{LA\_TA\_R}$ from Eq. 17, and $\delta$ is calculated as shown in Eq. 18, using Eq. 16.

$$\frac{I_{LA\_TA\_s}}{\eta_{s\_TA} E_{LA\_TA} T_{s\_TA} T_{O\_TA} F_{11}(f_{11} + g_{11})} = \frac{1}{2} \left[ 1 - a D_{s\_TA} \right] \tag{17}$$

$$\frac{I_{LA\_TA\_R}}{I_{LA\_TA\_T}} = \eta_{TA}\frac{1+a}{1-a} = \frac{\eta_{TA}}{\delta} \Rightarrow \delta = \frac{1}{\eta_{TA}}\frac{I_{LA\_TA\_T}}{I_{LA\_TA\_R}} \tag{18}$$

Due to the rotation of $HWP_{TA}$, the VLDR measurements cannot be acquired simultaneously with $F_{LA\_TA}$ (Eq. 9) and $\frac{I_{LB\_TA\_R}}{I_{LB\_TA\_T}}$ (Eq. A11), but they are acquired simultaneously with $F_{LA\_TB}$ (Eq. 10) and $\frac{I_{LB\_TB\_R}}{I_{LB\_TB\_T}}$ (Eq. A12).

## 7   First measurements

We present preliminary measurements from Athens, Greece, on June 15, 2021. The measurements were acquired during a dust-free day, with low aerosol optical depth (AOD) of 0.1 at 500 nm, as provided from the AERONET station in Athens. The absence of dust is supported from the WRF-Chem model simulations, indicating that desert dust has not been advected over the region, as well as from the low values of VLDR at 532 nm, measured with the PollyXT lidar of Antikythera (Baars et al., 2016), indicating spherical particles (not shown here). The instrument pointed at a viewing angle of $80^{o}$ off-zenith. Figure

13 shows the quicklooks of the attenuated backscatter signals from laser A ($I_{LA\_TA\_T}$, $I_{LA\_TA\_R}$, $I_{LA\_TB\_T}$ and $I_{LA\_TB\_R}$) acquired at $14:33-15:02$ UTC, along with the corresponding orientation flags ($F_{LA\_TA}$ and $F_{LA\_TB}$). During acquisition, the signals were averaged over 1 min. The signals from laser B are not shown, due its removal for repairs during that day. The dotted black line in Fig. 13 marks the full-overlap height of the signals, at 200 m above the ground (i.e., 400 m above sea level), as derived by the telecover test (Freudenthaler et al., 2018). The gaps in the quicklooks contain cloudy profiles which

are screened-out, resulting to a total duration of the cloud-free measurements of 22 min.

The aerosol layer extends up to $\sim 2.6$ km. Both orientation flags have values of $\sim 1$, showing the absence of oriented particles, as expected for a dust-free atmosphere. Specifically, the values shown in Fig. 13c) for $F_{LA\_TA}$ and 13f) for $F_{LA\_TB}$ are 1, with a standard deviation of $\pm 0.05$, up to 1.5 km where most of the aerosols reside (the standard deviation is calculated as the variation of the values of orientation flags from the full-overlap height up to 1.5 km). The standard deviation grows

larger at higher heights, and reaches $\pm 0.2$ at 2.6 km, at the top of the aerosol layer. Figure 14 shows the 22-min averages of the orientation flags $F_{LA\_TA}$ and $F_{LA\_TB}$ at $0.4-2.6$ km, from the overlap height, up to the top of the aerosol layer. The standard deviation of the 22-min average values of $F_{LA\_TA}$ and $F_{LA\_TB}$ is $\pm 0.02$ up to 1.5 km and grows up to $\pm 0.1$ at 2.6 km. The biases shown below 1 km, especially for $F_{LA\_TB}$, are not understood well yet, but they are within the standard deviation of $\pm 0.02$ at heights $1-1.5$ km.

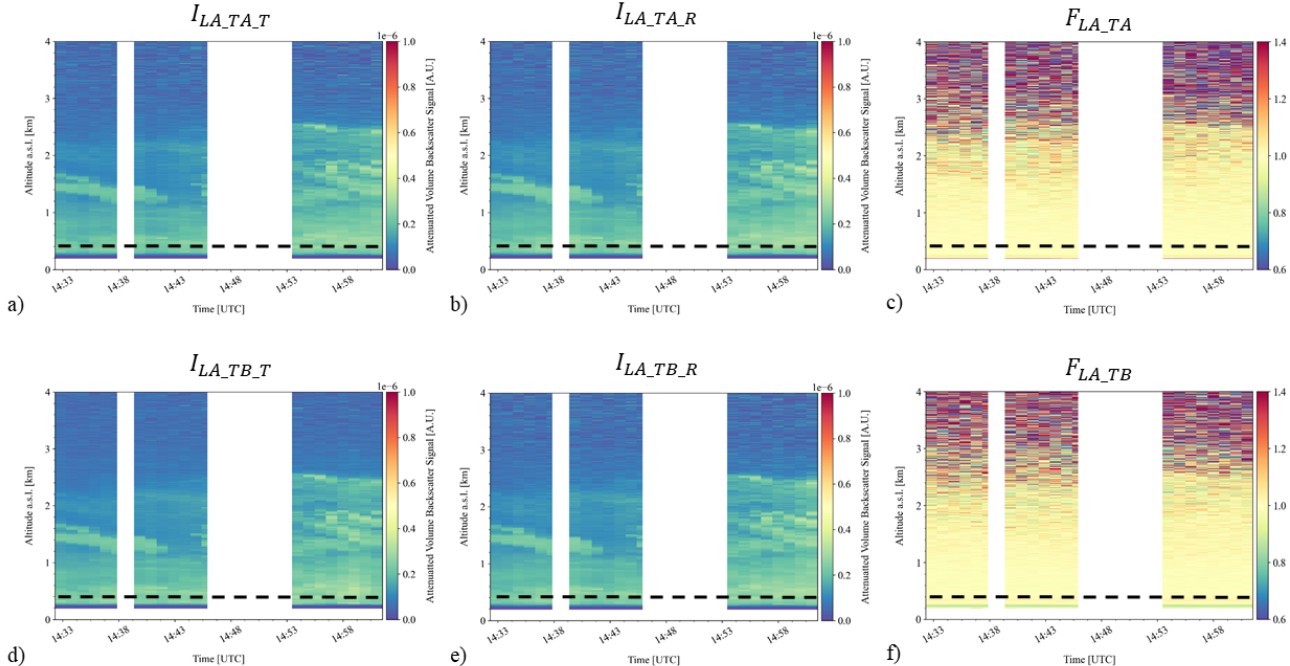

**Figure 13.** Lidar measurements at 1064nm, acquired at Athens, Greece, on June 15, 2021, at $14:33 - 15:02$ UTC. The viewing angle is at $80^o$ off-zenith. The height in the plots denotes the altitude and not the range of the measurements, and it is provided above sea level (a.s.l.). a) $I_{LA\_TA\_T}$ and b) $I_{LA\_TA\_R}$ are the attenuated backscatter signals from laser A measured at the detection unit of telescope A. d) $I_{LA\_TB\_T}$ and e) $I_{LA\_TB\_R}$ are the attenuated backscatter signals from laser A measured at the detection unit of telescope B. The corresponding orientation flags are shown in c) $F_{LA\_TA}$ and f) $F_{LA\_TB}$. The dotted black line shows the full-overlap height of the signals, at 200 m above the ground (i.e., 400 m above sea level). The gaps in the quicklooks contain cloudy profiles which are screened-out.

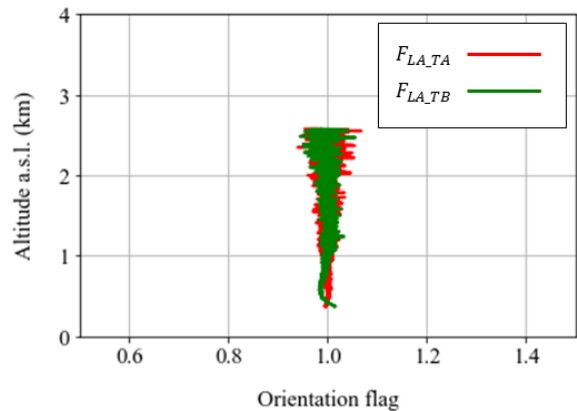

**Figure 14.** Average values of the orientation flags $F_{LA\_TA}$ (red) and $F_{LA\_TB}$ (green), for the measurements shown in Fig. 13, acquired in Athens, Greece, on June 15, 2021, at $14:33-15:02$ UTC. The averages are performed for the cloud-free profiles shown in Fig. 13, with duration of $22$ min, and they are plotted from the overlap height, up to the top of the aerosol layer, at $0.4-2.6$ km.

A measure of the quality of the signals is shown from the Rayleigh fit test (Freudenthaler et al., 2018) in Fig. 15. The signals agree well, with less than 10% difference, with the molecular atmosphere at the aerosol-free heights at $3-4$ km. This is not the case for altitudes $> 4$ km (not shown here), most probably due to the signal attenuation and the analog signal distortions. Due to the viewing angle of $80^o$ off-zenith, the altitude of $4$ km corresponds to $23$ km in range, thus the range of measurements for low AODs and for viewing angle of $80^o$ off-zenith, is $\sim 1-23$ km. For higher AODs (e.g. $0.4$ at $500$ nm) and zenith

measurements, this range reduces at $\sim 1-12$ km (not shown here).

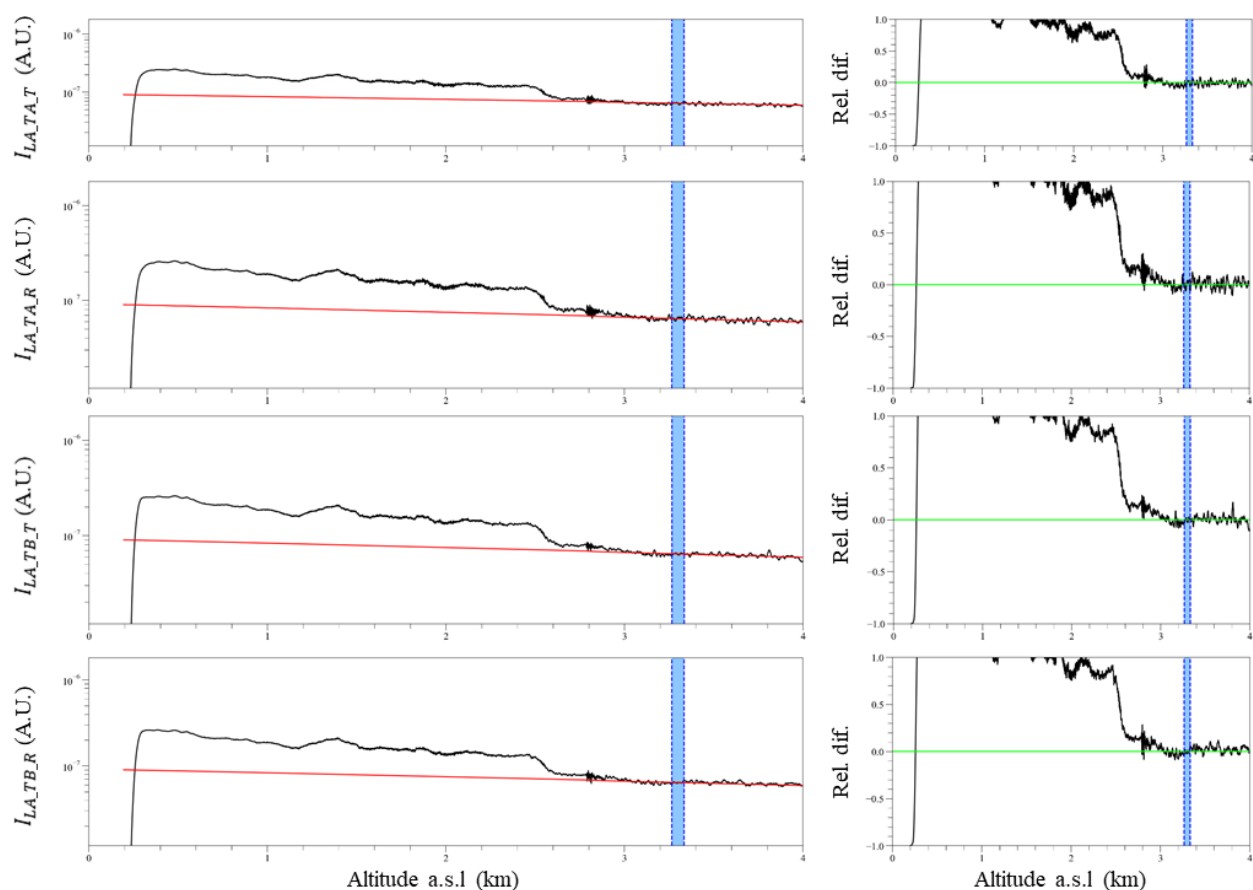

**Figure 15.** First column: The Rayleigh fit (Freudenthaler et al., 2018) of the averaged cloud-free signals in Fig. 13. Second column: the corresponding differences of the attenuated backscatter signals with the attenuated backscatter signals of the molecular atmosphere. The blue shaded area marks the reference height used for performing the Rayleigh fit.

## 8 Overview and future perspectives

The new polarization lidar nicknamed WALL-E is designed to monitor possible dust particle orientation in the Earth's atmosphere. This work describes in detail the conceptual design, its mechanical and optical parts, the calibration procedures, and finally the first, preliminary, measurements of the system.

The design extends the boundaries of lidar polarimetry, considering the various states of polarization emitted and detected by the system, and their interleaved emission and acquisition, which enables the detection of eight signals with two lasers/two telescopes/four detectors. Moreover, an important part of the design is the development of new methodologies for the calibration of the measurements and their alignment with the horizon, so as to define a reference system for the particle orientation. The mechanical developments include the compact design of the system, its mobility, its ability to perform measurements at the

field under a wide range of ambient conditions, and its ability to perform measurements at various zenith and azimuth viewing angles.

Further work is required towards improving the system performance and fulfill its objective. Firstly, the system has not yet measured dust orientation, or the orientation of other particles (e.g. rain as in Hayman et al. (2012)), thus it is not fully tested in this respect. The detection of the orientation of rain has been tried, but it hasn't been completed, since it entails high technical challenges, mainly due to the analog detection of the signals which are saturated from overlaying clouds and/or the rain. Although this is not impossible to cope, it requires extensive experimentation, which will be part of our future work. Moreover, further analysis should be done on the definition of the elliptical polarization of laser B, so as to maximize the information content provided in the measurements with respect to the dust particle microphysical properties. An extended analysis should be also performed for the characterization of the analog signal distortions, which are expected to affect the quality of the signals. Specifically, we will investigate their dependence on temperature and signal strength, and their change with height range and time. A very preliminary indication of their effect is shown in the first measurements presented herein, with a standard deviation of the measured orientation flags of $2 - 10\%$.

Resolving the scientific question of the role of dust electrification on dust removal processes, requires systematic observations of its main "tracer", the orientation of dust. Currently, the only indication of particle orientation comes from astronomical polarimetry measurements of dichroic extinction, which though cannot provide a strong proof for the phenomenon, due to their small number. Although polarimetric lidar and radar methods have been applied to quantitatively estimate the orientation of ice crystals, to our knowledge there is no such technique available to detect and characterize aerosol particle orientation. Our work envisages crossing the barriers by advancing the current lidar technologies towards developing a system able to detect and quantify particle orientation. The new polarization lidar presented herein will be capable to provide vertically-resolved measurements of dust orientation flags, along with measurements with information content on the microphysical properties of the dust particles.

If dust orientation is proven for the dust particles in the Earth's atmosphere, its detection and monitoring will unlock our ability for realistic simulations of the desert dust radiative impact, and optimize parameterizations for the natural aerosol component in Earth System Models in front of today's challenges posed by climate change. The applications of the new polarization lidar are not limited on this work but are anticipated to open new horizons for atmospheric remote sensing.

## Appendix A: Formulas of lidar signals $I_{i\_k\_s}$ and lidar signal ratios $\frac{I_{i\_k\_R}}{I_{i\_k\_T}}$

Equations A1-A2 show the analytic formulas of the signal $I_{i\_k\_s}$ (Eq. 4), for telescopes $k = TA$ and $TB$, considering ideal receiver optics (i.e. telescope, collimating lenses, bandpass filter), with no diattenuation, retardation and misalignment effects. Due to the rotation of the detection units after telescopes A and B, and the optical elements used for the calibration procedures as discussed in Section 4, Eq. A1-A2 are a simplified version of the actual equations of the signals $I_{i\_k\_s}$, which are provided in Eq. A3-A8. The analytic derivations of Eq. A3-A8 are provided in Section S2 of the Supplement. Finally, the signal ratios that are used instead of the individual signals due to calibration reasons, are provided in Eq. A9-A12.

## A1 Lidar signals $I_{i\_k\_s}$, considering no rotation of the detection units

$$\frac{I_{i\_TA\_s}}{\eta_{s\_TA}E_{i\_TA}T_{s\_TA}T_{O\_TA}F_{11}} = (f_{11}+g_{11})I_i + f_{12}D_{s\_TA}I_i + f_{21}Q_i + (f_{22}+g_{22})D_{s\_TA}Q_i +$$
$$+ f_{31}U_i + f_{32}D_{s\_TA}U_i + f_{41}V_i + f_{42}D_{s\_TA}V_i \tag{A1}$$

$$\frac{I_{i\_TB\_s}}{\eta_{s\_TB}E_{i\_TB}T_{s\_TB}T_{O\_TB}F_{11}} = I_i + (D_{s\_TB}c_{2\phi_{TB}}^2 I_i + Q_i)f_{12} + (-D_{s\_TB}c_{2\phi_{TB}}s_{2\phi_{TB}}I_i + U_i)f_{13} +$$
$$+ (-D_{s\_TB}s_{2\phi_{TB}}I_i + V_i)f_{14} + D_{s\_TB}c_{2\phi_{TB}}^2 Q_i f_{22} +$$
$$+ D_{s\_TB}(-c_{2\phi_{TB}}s_{2\phi_{TB}}Q_i + c_{2\phi_{TB}}^2 U_i)f_{23} + D_{s\_TB}(-s_{2\phi_{TB}}Q_i + c_{2\phi_{TB}}^2 V_i)f_{24} +$$
$$+ D_{s\_TB}c_{2\phi_{TB}}s_{2\phi_{TB}}U_i f_{33} + D_{s\_TB}(s_{2\phi_{TB}}U_i + c_{2\phi_{TB}}s_{2\phi_{TB}}V_i)f_{34} - D_{s\_TB}s_{2\phi_{TB}}V_i f_{44} +$$
$$+ I_i g_{11} + D_{s\_TB}c_{2\phi_{TB}}^2 Q_i g_{22} + D_{s\_TB}c_{2\phi_{TB}}s_{2\phi_{TB}}U_i g_{33} - D_{s\_TB}s_{2\phi_{TB}}V_i g_{44} \tag{A2}$$

$T_{s\_k}$ is the unpolarized transmittance ($s=T$) or reflectance ($s=R$) of the $PBS_k$, $T_{O\_k}$ is the transmittance of the receiver optics, $f_{ij}=\frac{F_{ij}}{F_{11}}$ and $g_{ij}=\frac{G_{ij}}{F_{11}}$ are the backscatter Stokes phase matrix elements of the oriented dust particles and gases, respectively, $I_i$, $Q_i$, $U_i$ and $V_i$ are the Stokes parameters of the light from the emission units of the lasers $i$, $D_{s\_k}$ are the diattenuation parameters of the transmitted or reflected channels of the $PBS_k$ followed by the cleaning polarizing sheet filters ($D_{T\_k}=1$ and $D_{R\_k}=-1$, respectively, see Section S1 in Supplement), $\phi_{TB}$ is the fast-axis-angle of the $QWP_{TB}$, and $c_{2\phi_{TB}}=cos(2\phi_{TB})$, $s_{2\phi_{TB}}=sin(2\phi_{TB})$.

## A2 Lidar signals $I_{i\_k\_s}$, considering the rotation of the detection units

Equations A3-A8 show the signals $I_{i\_k\_s}$, taking into account all the optical elements of the system, along with the rotation of the detection units after telescopes A and B, with respect to the frame coordinate system (Fig. 8). The analytic derivations of Eq. A3-A8 are provided in Section S2 of the Supplement.

### A2.1 $I_{LA\_TA\_s}$

$$\frac{I_{LA\_TA\_s}}{\eta_{s\_TA}E_{LA\_TA}T_{s\_TA}T_{O\_TA}F_{11}} = f_{11} + g_{11} + c_{4\vartheta_{LA}}f_{12} + s_{4\vartheta_{LA}}f_{13} +$$
$$+ D_{s\_TA}c_{(4\theta_{TA}+2\omega_{TA})}\big[f_{12} + c_{4\vartheta_{LA}}(f_{22}+g_{22}) + s_{4\vartheta_{LA}}f_{23}\big] +$$
$$+ D_{s\_TA}s_{(4\theta_{TA}+2\omega_{TA})}\big[-f_{13} - c_{4\vartheta_{LA}}f_{23} + s_{4\vartheta_{LA}}(f_{33}+g_{33})\big] \tag{A3}$$

After correcting $I_{LA\_TA\_s}$ for the rotation of the detection unit after telescope A, by setting the fast-axis-angle of $HWP_{TA}$ at $\theta_{TA}=-\frac{\omega_{TA}}{2}$ (Section 4.1), Eq. A3 is written as Eq. A4.

$$\frac{I_{LA\_TA\_s\_(\theta_{TA}=-\frac{\omega_{TA}}{2})}}{\eta_{s\_TA}E_{LA\_TA}T_{s\_TA}T_{O\_TA}F_{11}} = f_{11} + g_{11} + c_{4\vartheta_{LA}}f_{12} + s_{4\vartheta_{LA}}f_{13} + D_{s\_TA}\big[f_{12} + c_{4\vartheta_{LA}}(f_{22}+g_{22}) + s_{4\vartheta_{LA}}f_{23}\big] \tag{A4}$$

## A2.2 $I_{LB\_TA\_s}$

$$\frac{I_{LB\_TA\_s}}{\eta_{s\_TA}E_{LB\_TA}T_{s\_TA}T_{O\_TA}F_{11}} = f_{11} + \left[D_{s\_TA}c_{(4\theta_{TA}+2\omega_{TA})} + c_{2\varphi_{LB}}c_{(4\theta_{LB}-2\phi_{LB})}\right]f_{12} +$$

$$+ \left[c_{2\varphi_{LB}}s_{(4\theta_{LB}-2\phi_{LB})} - D_{s\_TA}s_{(4\theta_{TA}+2\omega_{TA})}\right]f_{13}$$

$$- s_{2\varphi_{LB}}f_{14} + D_{s\_TA}c_{(4\theta_{TA}+2\omega_{TA})}c_{2\varphi_{LB}}c_{(4\theta_{LB}-2\phi_{LB})}f_{22} +$$

$$+ D_{s\_TA}c_{2\varphi_{LB}}\left[c_{(4\theta_{TA}+2\omega_{TA})}s_{(4\theta_{LB}-2\phi_{LB})} - s_{(4\theta_{TA}+2\omega_{TA})}c_{(4\theta_{LB}-2\phi_{LB})}\right]f_{23} + \quad \text{(A5)}$$

$$- D_{s\_TA}c_{(4\theta_{TA}+2\omega_{TA})}s_{2\varphi_{LB}}f_{24} + D_{s\_TA}s_{(4\theta_{TA}+2\omega_{TA})}c_{2\varphi_{LB}}s_{(4\theta_{LB}-2\phi_{LB})}f_{33} +$$

$$- D_{s\_TA}s_{(4\theta_{TA}+2\omega_{TA})}s_{2\varphi_{LB}}f_{34} +$$

$$+ g_{11} + D_{s\_TA}c_{(4\theta_{TA}+2\omega_{TA})}c_{2\varphi_{LB}}c_{(4\theta_{LB}-2\phi_{LB})}g_{22} +$$

$$+ D_{s\_TA}s_{(4\theta_{TA}+2\omega_{TA})}c_{2\varphi_{LB}}s_{(4\theta_{LB}-2\phi_{LB})}g_{33}$$

After correcting $I_{LB\_TA\_s}$ for the rotation of the detection unit after telescope A, by setting the fast-axis-angle of $HWP_{TA}$ at $\theta_{TA} = -\frac{\omega_{TA}}{2}$ (Section 4.1), Eq. A5 is written as Eq. A6.

$$\frac{I_{LB\_TA\_s\_(\theta_{TA}=-\frac{\omega_{TA}}{2})}}{\eta_{s\_TA}E_{LB\_TA}T_{s\_TA}T_{O\_TA}F_{11}} = f_{11} + \left[D_{s\_TA} + c_{2\varphi_{LB}}c_{(4\theta_{LB}-2\phi_{LB})}\right]f_{12} + c_{2\varphi_{LB}}s_{(4\theta_{LB}-2\phi_{LB})}f_{13} - s_{2\varphi_{LB}}f_{14} +$$

$$+ D_{s\_TA}c_{2\varphi_{LB}}c_{(4\theta_{LB}-2\phi_{LB})}f_{22} + D_{s\_TA}c_{2\varphi_{LB}}s_{(4\theta_{LB}-2\phi_{LB})}f_{23} - D_{s\_TA}s_{2\varphi_{LB}}f_{24} +$$

$$+ g_{11} + D_{s\_TA}c_{2\varphi_{LB}}c_{(4\theta_{LB}-2\phi_{LB})}g_{22}$$

435

$$\text{(A6)}$$

## A2.3 $I_{LA\_TB\_s}$

$$\frac{I_{LA\_TB\_s}}{\eta_{s\_TB}E_{LA\_TB}T_{s\_TB}T_{O\_TB}F_{11}} = f_{11} + c_{4\vartheta_{LA}}f_{12} + s_{4\vartheta_{LA}}f_{13} + D_{s\_TB}f_{14} + D_{s\_TB}c_{4\vartheta_{LA}}f_{24} - D_{s\_TB}s_{4\vartheta_{LA}}f_{34} + g_{11}$$

$$\text{(A7)}$$

## A2.4 $I_{LB\_TB\_s}$

$$\frac{I_{LB\_TB\_s}}{\eta_{s\_TB}E_{LB\_TB}T_{s\_TB}T_{O\_TB}F_{11}} = f_{11} + c_{2\varphi_{LB}}c_{(4\theta_{LB}-2\phi_{LB})}f_{12} + c_{2\varphi_{LB}}s_{(4\theta_{LB}-2\phi_{LB})}f_{13} + \left[D_{s\_TB} - s_{2\varphi_{LB}}\right]f_{14} +$$

$$+ D_{s\_TB}c_{2\varphi_{LB}}c_{(4\theta_{LB}-2\phi_{LB})}f_{24} - D_{s\_TB}c_{2\varphi_{LB}}s_{(4\theta_{LB}-2\phi_{LB})}f_{34} - D_{s\_TB}s_{2\varphi_{LB}}f_{44} +$$

$$+ g_{11} - D_{s\_TB}s_{2\varphi_{LB}}g_{44}$$

$$\text{(A8)}$$

## A3   Lidar signal ratios $\frac{I_{i\_k\_R}}{I_{i\_k\_T}}$, after correcting the rotation of the detection units

For calibration reasons, we use the ratios of the reflected and transmitted channels, instead of the individual signals. The signal ratios of lasers A and B, measured at the detection units after telescopes A and B, are provided in Eq. A9-A12.

$$\frac{I_{LA\_TA\_R}}{I_{LA\_TA\_T}} = \eta_{TA}\frac{1 - f_{12} + f_{13} - f_{23} + g_{11}}{1 + f_{12} + f_{13} + f_{23} + g_{11}} \tag{A9}$$

$$\frac{I_{LA\_TB\_R}}{I_{LA\_TB\_T}} = \eta_{TB}\frac{1 + f_{13} - f_{14} + f_{34} + g_{11}}{1 + f_{13} + f_{14} - f_{34} + g_{11}} \tag{A10}$$

$$\frac{I_{LB\_TA\_R}}{I_{LB\_TA\_T}} = \eta_{TA}\frac{1 + 0.15f_{12} + 0.17f_{13} + 0.5f_{14} - 0.85f_{22} - 0.17f_{23} - 0.5f_{24} + g_{11} - 0.85g_{22}}{1 + 1.85f_{12} + 0.17f_{13} + 0.5f_{14} + 0.85f_{22} + 0.17f_{23} + 0.5f_{24} + g_{11} + 0.85g_{22}} \tag{A11}$$

$$\frac{I_{LB\_TB\_R}}{I_{LB\_TB\_T}} = \eta_{TB}\frac{1 + 0.85f_{12} + 0.17f_{13} - 0.5f_{14} - 0.85f_{24} + 0.17f_{34} - 0.5f_{44} + g_{11} - 0.5g_{44}}{1 + 0.85f_{12} + 0.17f_{13} + 1.5f_{14} + 0.85f_{24} - 0.17f_{34} + 0.5f_{44} + g_{11} + 0.5g_{44}} \tag{A12}$$

## Appendix B:  Derivation of $\varepsilon_{TA}$ for correcting $I_{LA\_TA\_s}$ and $I_{LB\_TA\_s}$, for the rotation of the detection unit after telescope A.

After placing a linear polarizer in front of the window of laser A at $45^o$ from $x_F$-axis (Fig. 9), we acquire the measurements

$I_{LA\_TA\_s\_45^o}$ (Eq. S24 in the Supplement). Considering $\theta_{TA} = \frac{-\omega_{TA}}{2} + \varepsilon_{TA}$ we derive $I_{LA\_TA\_S\_45^o\_(\theta_{TA}=-\frac{\omega_{TA}}{2}+\varepsilon_{TA})}$ as shown in Eq. B1.

$$\frac{I_{LA\_TA\_s\_45^o\_(\theta_{TA}=-\frac{\omega_{TA}}{2}+\varepsilon_{TA})}}{\eta_{s\_TA}E_{LA\_TA}T_{s\_TA}T_{O\_TA}F_{11}} = \frac{1}{2}\Big[f_{11} + g_{11} + D_{s\_TA}s_{(4\theta_{TA}+2\omega_{TA})}(f_{33} + g_{33})\Big](1 + s_{4\vartheta_{LA}}) =$$

$$= \frac{1}{2}\Big[f_{11} + g_{11} + D_{s\_TA}s_{(4(-\frac{\omega_{TA}}{2}+\varepsilon_{TA})+2\omega_{TA})}(f_{33} + g_{33})\Big](1 + s_{4\vartheta_{LA}}) =$$

$$= \frac{1}{2}\Big[f_{11} + g_{11} + D_{s\_TA}s_{4\varepsilon_{TA}}(f_{33} + g_{33})\Big](1 + s_{4\vartheta_{LA}}) \Rightarrow \tag{B1}$$

$$\Rightarrow \frac{I_{LA\_TA\_s\_45^o\_(\theta_{TA}=-\frac{\omega_{TA}}{2}+\varepsilon_{TA})}}{\eta_{s\_TA}E_{LA\_TA}T_{s\_TA}T_{O\_TA}F_{11}(f_{11} + g_{11})} == \frac{1}{2}\Big[1 - aD_{s\_TA}s_{4\varepsilon_{TA}}\Big](1 + s_{4\vartheta_{LA}})$$

$a$ is the atmospheric polarization parameter with $a = \frac{f_{22}+g_{22}}{f_{11}+g_{11}} = -\frac{f_{33}+g_{33}}{f_{11}+g_{11}}$.

We derive $\varepsilon_{TA}$ in a similar way as the "$\Delta 90^o$ calibration" (Section 11 in Freudenthaler (2016)), by rotating the $HWP_{TA}$ by an additional angle of $0^o$ and $45^o$ with respect to the $x_{DU_{TA}}$-axis (Fig. 10). The respective calculations are provided in Eq. B2-B9:

$$\eta^*\left(\theta_{TA} = -\frac{\omega_{TA}}{2} + \varepsilon_{TA}\right) = \frac{I_{LA\_TA\_R\_45^o\_(\theta_{TA}=-\frac{\omega_{TA}}{2}+\varepsilon_{TA})}}{I_{LA\_TA\_T\_45^o(\theta_{TA}=-\frac{\omega_{TA}}{2}+\varepsilon_{TA})}} = \eta_{TA}\frac{\left(1+as_{4\varepsilon_{TA}}\right)\left(1+s_{4\vartheta_{LA}}\right)}{\left(1-as_{4\varepsilon_{TA}}\right)\left(1+s_{4\vartheta_{LA}}\right)} = \eta_{TA}\frac{1+as_{4\varepsilon_{TA}}}{1-as_{4\varepsilon_{TA}}} \quad \text{(B2)}$$

$$\eta^*\left(\theta_{TA} = +45^o - \frac{\omega_{TA}}{2} + \varepsilon_{TA}\right) = \frac{I_{LA\_TA\_R\_45^o\_(\theta_{TA}=45-\frac{\omega_{TA}}{2}+\varepsilon_{TA})}}{I_{LA\_TA\_T\_45^o(\theta_{TA}=45-\frac{\omega_{TA}}{2}+\varepsilon_{TA})}} = \eta_{TA}\frac{\left(1+as_{4(45+\varepsilon_{TA})}\right)\left(1+s_{4\vartheta_{LA}}\right)}{\left(1-as_{4(45+\varepsilon_{TA})}\right)\left(1+s_{4\vartheta_{LA}}\right)} =$$

$$\quad \text{(B3)}$$

$$= \eta_{TA}\frac{1-as_{4\varepsilon_{TA}}}{1+as_{4\varepsilon_{TA}}}$$

$$Y(\varepsilon_{TA},a) = \frac{\eta^*\left(\theta_{TA} = -\frac{\omega_{TA}}{2} + \varepsilon_{TA}\right) - \eta^*\left(\theta_{TA} = +45^o - \frac{\omega_{TA}}{2} + \varepsilon_{TA}\right)}{\eta^*\left(\theta_{TA} = -\frac{\omega_{TA}}{2} + \varepsilon_{TA}\right) + \eta^*\left(\theta_{TA} = +45^o - \frac{\omega_{TA}}{2} + \varepsilon_{TA}\right)} = \frac{2as_{4\varepsilon_{TA}}}{1+a^2 s_{4\varepsilon_{TA}}^2} \quad \text{(B4)}$$

Following the tangent half-angle substitution (Section S.12.1 in Freudenthaler (2016)) we derive $\varepsilon_{TA}$ and $a$ by successive approximation, as shown in Eq. B5-B9.

$$\varepsilon_{TA} = \frac{1}{4}arcsin\left[\frac{1}{a}tan\left(\frac{arcsin\left(Y(\varepsilon_{TA},a)\right)}{2}\right)\right] \quad \text{(B5)}$$

$$a = \frac{1}{s_{4\varepsilon_{TA}}}tan\left(\frac{arcsin\left(Y(\varepsilon_{TA},a)\right)}{2}\right) \quad \text{(B6)}$$

As a first approximation of $\varepsilon_{TA}$ we calculate $\varepsilon_{TA_l}$ with Eq. B7.

$$\varepsilon_{TA_l} = \frac{1}{4}arcsin\left[tan\left(\frac{arcsin\left(Y(\varepsilon_{TA},a)\right)}{2}\right)\right] < \varepsilon_{TA} \quad \text{(B7)}$$

After adjusting the $HWP_{TA}$ by $-\varepsilon_{TA_l}$, which results in $\theta_{TA} = -\frac{\omega_{TA}}{2} + \varepsilon_{TA} - \varepsilon_{TA_l}$, we derive $Y(\varepsilon_{TA} - \varepsilon_{TA_l}, a)$ (Eq. B8). Then, $\varepsilon_{TA}$ is calculated by Eq. B9.

$$Y(\varepsilon - \varepsilon_{TA_l}, a) = \frac{2as_{4(\varepsilon - \varepsilon_{TA_l})}}{1 + a^2 s^2_{4(\varepsilon - \varepsilon_{TA_l})}} \tag{B8}$$

$$\frac{Y(\varepsilon_{TA} - \varepsilon_{TA_l}, a)}{Y(\varepsilon_{TA}, a)} = \frac{s_{4(\varepsilon_{TA} - \varepsilon_{TA_l})}(1 + s^2_{4\varepsilon_{TA}} a^2)}{s_{4\varepsilon_{TA}}(1 + s^2_{4(\varepsilon_{TA} - \varepsilon_{TA_l})} a^2)} \approx \frac{s_{4(\varepsilon_{TA} - \varepsilon_{TA_l})}}{s_{4\varepsilon_{TA}}} \approx \frac{\varepsilon_{TA} - \varepsilon_{TA_l}}{\varepsilon_{TA}} = 1 - \frac{\varepsilon_{TA_l}}{\varepsilon_{TA}} \Rightarrow$$

$$\tag{B9}$$

$$\Rightarrow \varepsilon_{TA} \approx \frac{Y(\varepsilon_{TA}, a)}{Y(\varepsilon_{TA}, a) - Y(\varepsilon_{TA} - \varepsilon_{TA_l}, a)} \varepsilon_{TA_l}$$

Figure B1 shows the test performed for zeroing $\varepsilon_{TA}$ and setting $\theta_{TA} = \frac{-\omega_{TA}}{2}$ (step 3 in Fig. 10), by the successive steps described above. Specifically, with zero rotation of the $HWP_{TA}$, $\varepsilon_{TA_l} = 4.69^o$, with $-4.69^o$ rotation of the $HWP_{TA}$, $\varepsilon_{TA_l} = -0.22^o$, and finally with $-4.69^o + 0.22^o = -4.47^o$ rotation of the $HWP_{TA}$, $\varepsilon_{TA_l} = 0^o$.

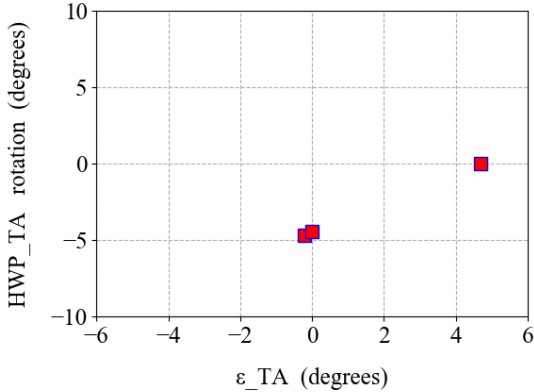

**Figure B1.** The test performed for zeroing $\varepsilon_{TA}$ and setting of $\theta_{TA} = \frac{-\omega_{TA}}{2}$, for the correction of $\mathbf{I_{LA\_TA\_s}}$ and $\mathbf{I_{LB\_TA\_s}}$, for the rotation of the detection unit after telescope A.

## Appendix C: Derivation of $\varepsilon_{LA}$ for defining the polarization of $i_{LA}$ with respect to the horizon

For the derivation of the $\varepsilon_{LA}$ we use the signals $I_{LA\_TA\_s}$ (Eq. A4), after they are corrected for the effect of the rotation of the

detection unit after telescope A, by setting $\theta_{TA} = -\frac{\omega_{TA}}{2}$ (Section 4.1). Moreover, we consider that the atmosphere consists of

randomly-oriented particles, thus the off-diagonal elements in Eq. A4 are zero. Then, the signals $I_{LA\_TA\_S}$ are provided in Eq. C1.

$$I_{LA\_TA\_S} = \eta_{s\_TA} E_{LA\_TA} T_{s\_TA} T_{O\_TA} F_{11} \left[ f_{11} + g_{11} + D_{s\_TA} c_{4\vartheta_{LA}} (f_{22} + g_{22}) \right] =$$
$$= \eta_{s\_TA} E_{LA\_TA} T_{s\_TA} T_{O\_TA} F_{11} (f_{11} + g_{11}) \left[ 1 + a D_{s\_TA} c_{4\vartheta_{LA}} \right] \tag{C1}$$

Where, $\vartheta_{LA} = \varepsilon_{LA}$ and $a$ is the atmospheric polarization parameter, $a = \frac{f_{22} + g_{22}}{f_{11} + g_{11}}$.

We rotate the $HWP_{LA}$ by $+22.5^o$ and $-22.5^o$ with respect to the $x_F$-axis and we derive $\varepsilon_{LA}$ by performing the "$\Delta 90^o$ calibration" of Freudenthaler (2016), as shown in Eq. C2-C8.

$$\eta^* (\vartheta_{LA} = +22.5^o + \varepsilon_{LA}) = \frac{I_{LA\_TA\_R(\vartheta_{LA} = 22.5 + \varepsilon_{LA})}}{I_{LA\_TA\_T(\vartheta_{LA} = 22.5 + \varepsilon_{LA})}} = \eta_{TA} \frac{1 - a c_{4(22.5 + \varepsilon_{LA})}}{1 + a c_{4(22.5 + \varepsilon_{LA})}} =$$

$$= \eta_{TA} \frac{1 - a c_{(90 + 4\varepsilon_{LA})}}{1 + a c_{(90 + 4\varepsilon_{LA})}} = \eta_{TA} \frac{1 + a s_{4\varepsilon_{LA}}}{1 - a s_{4\varepsilon_{LA}}} \tag{C2}$$

$$\eta^* (\vartheta_{LA} = -22.5^o + \varepsilon_{LA}) = \frac{I_{LA\_TA\_R(\vartheta_{LA} = -22.5 + \varepsilon_{LA})}}{I_{LA\_TA\_T(\vartheta_{LA} = -22.5 + \varepsilon_{LA})}} = \eta_{TA} \frac{1 - a c_{(-90 + 4\varepsilon_{LA})}}{1 + a c_{(-90 + 4\varepsilon_{LA})}} =$$

$$= \eta_{TA} \frac{1 - a s_{4\varepsilon_{LA}}}{1 + a s_{4\varepsilon_{LA}}} \tag{C3}$$

$$Y(\varepsilon_{LA}, a) = \frac{\eta^* (\vartheta_{LA} = +22.5^o + \varepsilon_{LA}) - \eta^* (\vartheta_{LA} = -22.5^o + \varepsilon_{LA})}{\eta^* (\vartheta_{LA} = +22.5^o + \varepsilon_{LA}) + \eta^* (\vartheta_{LA} = -22.5^o + \varepsilon_{LA})} = \frac{2 a s_{4\varepsilon_{LA}}}{1 + a^2 s_{4\varepsilon_{LA}}^2} \tag{C4}$$

Following the tangent half-angle substitution (Section S.12.1 in Freudenthaler (2016)) we derive $\varepsilon_{LA}$, as shown in Eq. C5-C8.

$$\varepsilon_{LA} = \frac{1}{4} arcsin \left[ \frac{1}{a} tan \left( \frac{arcsin (Y(\varepsilon_{LA}, a))}{2} \right) \right] \tag{C5}$$

As a first approximation of $\varepsilon_{LA}$ we calculate $\varepsilon_{LA_l}$ with Eq. C6.

$$\varepsilon_{LA_l} = \frac{1}{4}arcsin\left[tan\left(\frac{arcsin\left(Y(\varepsilon_{LA},a)\right)}{2}\right)\right] < \varepsilon_{LA} \tag{C6}$$

After adjusting the $HWP_{LA}$ by $-\varepsilon_{LA_l}$, which results in $\vartheta_{LA} = \varepsilon_{LA} - \varepsilon_{LA_l}$ with respect to the $x_F$-axis, we derive $Y(\varepsilon_{LA} - \varepsilon_{LA_l},a)$ (Eq. C7). Then, $\varepsilon_{LA}$ is calculated by Eq. C8.

$$Y(\varepsilon_{LA} - \varepsilon_{LA_l},a) = \frac{2as_{4(\varepsilon_{LA}-\varepsilon_{LA_l})}}{1 + a^2 s^2_{4(\varepsilon_{LA}-\varepsilon_{LA_l})}} \tag{C7}$$

$$\frac{Y(\varepsilon_{LA} - \varepsilon_{LA_l},a)}{Y(\varepsilon_{LA},a)} = \frac{s_{4(\varepsilon_{LA}-\varepsilon_{LA_l})}(1 + s^2_{4\varepsilon_{LA}}a^2)}{s_{4\varepsilon_{LA}}(1 + s^2_{4(\varepsilon_{LA}-\varepsilon_{LA_l})}a^2)} \approx \frac{s_{4(\varepsilon_{LA}-\varepsilon_{LA_l})}}{s_{4\varepsilon_{LA}}} \approx \frac{\varepsilon_{LA} - \varepsilon_{LA_l}}{\varepsilon_{LA}} = 1 - \frac{\varepsilon_{LA_l}}{\varepsilon_{LA}} \Rightarrow$$

$$\tag{C8}$$

$$\Rightarrow \varepsilon_{LA} \approx \frac{Y(\varepsilon_{LA},a)}{Y(\varepsilon_{LA},a) - Y(\varepsilon_{LA} - \varepsilon_{LA_l},a)}\varepsilon_{LA_l}$$

Figure C1 shows the test performed for zeroing $\varepsilon_{LA}$, by the successive steps described above. Specifically, with zero rotation of the $HWP_{LA}$, $\varepsilon_{LA_l} = 4.03^o$, with $-4.03^o$ rotation of the $HWP_{LA}$, $\varepsilon_{LA_l} = 0.56^o$, with $-4.03^o - 0.56^o = -4.59^o$ rotation of the $HWP_{LA}$, $\varepsilon_{LA_l} = 0.13^o$, and finally with $-4.59^o - 0.13^o = -4.72^o$ rotation of the $HWP_{LA}$, $\varepsilon_{LA_l} = 0^o$.

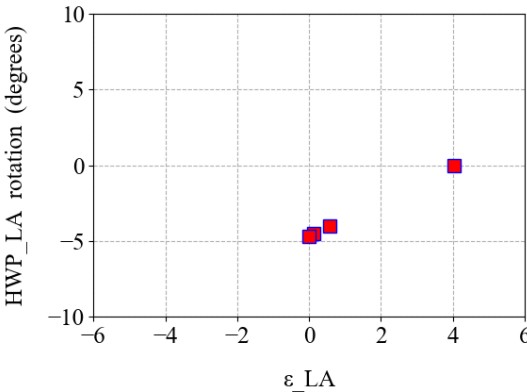

**Figure C1.** The test performed for zeroing $\varepsilon_{LA}$, for defining the polarization of $i_{LA}$ with respect to the horizon.

**Appendix D: Derivation of the angle of the polarization ellipse $\alpha$ and of the degree of linear polarization $b$, for defining the polarization of $i_{LB}$ with respect to the horizon.**

**D1  Derivation of the angle of the polarization ellipse $\alpha$**

The signals $I_{LB\_TA\_s}$ (Eq. A6) are provided by Eq. D1, considering an atmosphere with randomly-oriented particles (all off-diagonal elements of the backscatter matrix are zero), and that the detection unit after telescope A is aligned with the system frame (Section 4.1). Note that in Eq. D1 "$a$" is the atmospheric polarization parameter, whereas "$\alpha$" is the the angle of the polarization ellipse of the light from the emission unit of laser B.

$$\frac{I_{LB\_TA\_s}}{\eta_{s\_TA}E_{LB\_TA}T_{s\_TA}T_{O\_TA}F_{11}(f_{11}+g_{11})} = 1 + D_{s\_TA}c_{2\varphi_{LB}}c_{(4\theta_{LB}-2\phi_{LB})}\frac{f_{22}+g_{22}}{f_{11}+g_{11}} = 1 + abD_{s\_TA}c_{2\alpha} \tag{D1}$$

In order to derive the angle $\alpha$ we rotate the $HWP_{LB}$ by $+22.5^o$ and $-22.5^o$ with respect to the $x_F$-axis, so as $\alpha_{+45^o} = \alpha + 45^o$ and $\alpha_{-45^o} = \alpha - 45^o$, respectively (since $\alpha = 2\theta_{LB} - \phi_{LB}$, Eq. 13). We then perform a methodology similar to the "$\Delta 90^o$ calibration" of Freudenthaler (2016) to derive $\alpha$, as shown in Eq. D2-D8.

$$\eta^*(\alpha_{+45^o} = \alpha + 45^o) = \frac{I_{LB\_TA\_R\_\alpha_{+45^o}}}{I_{LB\_TA\_T\_\alpha_{+45^o}}} = \eta_{TA}\frac{1 - abc_{(2\alpha+90)}}{1 + abc_{(2\alpha+90)}} = \eta_{TA}\frac{1 + abs_{2\alpha}}{1 - abs_{2\alpha}} \tag{D2}$$

$$\eta^*(\alpha_{-45^o} = \alpha - 45^o) = \frac{I_{LB\_TA\_R\_\alpha_{-45^o}}}{I_{LB\_TA\_T\_\alpha_{-45^o}}} = \eta_{TA}\frac{1 - abc_{(2\alpha-90)}}{1 + abc_{(2\alpha-90)}} = \eta_{TA}\frac{1 - abs_{2\alpha}}{1 + abs_{2\alpha}} \tag{D3}$$

$$Y(\alpha,a,b) = \frac{\eta^*(\alpha_{+45^o} = \alpha + 45^o) - \eta^*(\alpha_{-45^o} = \alpha - 45^o)}{\eta^*(\alpha_{+45^o} = \alpha + 45^o) + \eta^*(\alpha_{-45^o} = \alpha - 45^o)} = \frac{2abs_{2\alpha}}{1 + a^2b^2s_{2\alpha}^2} \tag{D4}$$

Following the tangent half-angle substitution (Section S.12.1 in Freudenthaler (2016)) we derive $\alpha$ by successive approximation, as shown in Eq. D5-D9.

$$\alpha = \frac{1}{2}arcsin\left[\frac{1}{ab}tan\left(\frac{arcsin\left(Y(\alpha,a,b)\right)}{2}\right)\right] \tag{D5}$$

As a first approximation of $\alpha$ we calculate $\alpha_l$ with Eq. D6.

$$\alpha_l = \frac{1}{2}arcsin\left[tan\left(\frac{arcsin\left(Y(\alpha,a,b)\right)}{2}\right)\right] \tag{D6}$$

After adjusting the $HWP_{LB}$ by $-\frac{\alpha_l}{2}$, which results in $\alpha_{new} = \alpha - \alpha_l$ (Eq. 13), we derive $Y(\alpha - \alpha_l, a, b)$ (Eq. D7). Then, $\alpha$ is calculated by Eq. D8.

$$Y(\alpha - \alpha_l, a, b) = \frac{\eta^*(+45^o + \alpha - \alpha_l) - \eta^*(-45^o + \alpha - \alpha_l)}{\eta^*(+45^o + \alpha - \alpha_l) + \eta^*(-45^o + \alpha - \alpha_l)} == \frac{2abs_{2(\alpha - \alpha_l)}}{1 + a^2 b^2 s_{2(\alpha - \alpha_l)}^2} \tag{D7}$$

$$\frac{Y(\alpha - \alpha_l, a, b)}{Y(\alpha, a, b)} = \frac{s_{2(\alpha - \alpha_l)}\left(1 + a^2 b^2 s_{2\alpha}^2\right)}{s_{2\alpha}\left(1 + a^2 b^2 s_{2(\alpha - \alpha_l)}^2\right)} \approx \frac{\alpha - \alpha_l}{\alpha} = 1 - \frac{\alpha_l}{\alpha} \Rightarrow \tag{D8}$$

$$\Rightarrow \alpha \approx \frac{Y(\alpha, a, b)}{Y(\alpha, a, b) - Y(\alpha - \alpha_l, a, b)} \alpha_l$$

## D2 Derivation of the degree of linear polarization $b$

As shown in Fig. 12, after calculating $\alpha$ in Section D1, we rotate the $HWP_{LB}$ by $-\frac{\alpha}{2}$ and by $22.5^o - \frac{\alpha}{2}$, so as $\alpha_1 = 0^o$ and $\alpha_2 = 45^o$ with respect to the $x_F$-axis, respectively (Eq. 13).

Then, $b$ is derived from Eq. D9-D11 (using Eq. D1), considering that we have already derived the atmospheric polarization parameter $a$ using the measurements of laser A at the detection unit after telescope A, as shown in Section 6.

$$\eta^*(\alpha_1 = 0^o) = \frac{I_{LB\_TA\_R\_\alpha_1}}{I_{LB\_TA\_T\_\alpha_1}} = \eta_{TA}\frac{1 - abc_0}{1 + abc_0} = \eta_{TA}\frac{1 - ab}{1 + ab} \tag{D9}$$

$$\eta^*(\alpha_2 = +45^o) = \frac{I_{LB\_TA\_R\_\alpha_2}}{I_{LB\_TA\_T\_\alpha_2}} = \eta_{TA}\frac{1 - abc_{90}}{1 + abc_{90}} = \eta_{TA} \tag{D10}$$

$$b = \frac{1}{a}\frac{\eta^*(\alpha_2 = +45^o) - \eta^*(\alpha_1 = 0^o)}{\eta^*(\alpha_2 = +45^o) + \eta^*(\alpha_1 = 0^o)} \tag{D11}$$

*Author contributions.* AT and VF formulated the measurement strategy and the emission and detection design of the system, along with the calibration procedures. VF conceived the "two-laser/two-telescope" concept. GG and AL developed the optomechanical design of the

instrument. AT and SM tested and optimized the instrument, and acquired the measurements shown herein, with the support of AL, GG, GT, GD, CE and VF. AT, GD and GG performed simulations for defining the capabilities of the system in terms of SNR, JG provided guidance for the scattering calculations used in these simulations and TG provided the corresponding technical support. PP, NS and IB provided software for the quality assurance tests of the measurements. VA supervised and directed the whole project. AT wrote the manuscript and VF, VA, IB, GG, AL, JG and PP provided corrections and suggestions.

*Competing interests.* No competing interests are present.

*Acknowledgements.* The work is supported by the European Research Council under the European Community's Horizon 2020 research and innovation framework program/ERC grant agreement 725698 (D-TECT). We acknowledge PRACE for awarding us access to MareNostrum at Barcelona Supercomputing Center (BSC), Spain. This work was supported by computational time granted from the National Infrastructures for Research and Technology S.A. (GRNET S.A.) in the National HPC facility - ARIS - under project ID pa170906-ADDAPAS, pr005038-
REMOD, pr007032-RApID, pr009019-EXEED and pr011016-DSA. Part of the work performed for this study was funded by Romanian National Core Program Contract No.18N/2019 and by the European Regional Development Fund through Competitiveness Operational Programme 2014–2020, POC-A.1-A.1.1.1- F- 2015, project Research Centre for environment and Earth Observation CEO-Terra.

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
