# Peer review of "Polarization lidar for detecting dust orientation: System design and calibration."

_Atmospheric Measurement Techniques, 2021_

## Referee Comment (RC1)

Comments on Notation and Equations (the authors are not obligated to address this):
The polarization calculations in this manuscript are cumbersome and difficult to follow. For example, Eq. 8 has 65 terms (not all unique). Using linear algebra, it should be possible to represent the operation of this instrument in a way that's more intuitive.

Assuming a linear input polarization, it is possible to obtain any *linear* polarization state using a HWP. It is also possible to obtain *all* polarization states on the Poincare sphere with the combination of a HWP and a QWP. This is in accordance with the instrument design. So in the approximation that no other optic in the system is affecting polarization after the wave plates, there is no reason to belabor the definitions of the output polarization or the waveplate Mueller matrices and angles. All the cosines and sines in Eq. 8 are unnecessary. One can easily write the output state of laser A as

$$\mathbf{S}_A = \begin{bmatrix} S_1 & S_2 & S_3 & 0 \end{bmatrix}^T$$

where $S_3 = \sqrt{S_1^2 - S_2^2}$

And laser B as

$$\mathbf{S}_B = \begin{bmatrix} S_1 & S_2 & S_3 & S_4 \end{bmatrix}^T$$

where $S_4 = \sqrt{S_1^2 - S_2^2 - S_3^2}$

As part of the optimization process, we need to determine these vectors. (Note that the elements of these two vectors are not the same and that the degrees of freedom can be further described by the Poincare sphere polar coordinates or normalization).

By a very similar argument, you can fully evaluate the Mueller matrices in the receiver equations with the output vector **e** and describe each channel with a single vector that is the diattenuation vector of the receiver channel. The result is a row vector, analogous to a the transmitted Stokes vector, describing the channel's preference to transmit a particular polarization. Like with the transmitted polarization states, this diattenuation vector can take the form of any *linear* polarization if a linear analyzer is preceded by a HWP. The diattenuation vector can take the form of *any* polarization if a linear analyzer is preceded by a combination HWP and QWP. So now you have the receiver taking the form

$$\mathbf{D}_C = \begin{bmatrix} D_1 & D_2 & D_3 & 0 \end{bmatrix}$$

And

$$\mathbf{D}_D = \begin{bmatrix} D_1 & D_2 & D_3 & D_4 \end{bmatrix}$$

Instead of expanding Mueller matrices of individual polarization elements, why not do what Kaul 2004 and Hayman 2012 did and write that an optical measurement is described by a vector projection of the scattering phase matrix with a measurement described by the combination of the transmitted and detected states?

$$I = \mathbf{p}^T(\mathbf{f} + \mathbf{g})$$

Where **f** and **g** are vectorized scattering matrices.

The intensity measured on the detector is the projection of the scattering matrix via an incident and detected polarization state. The elements of that projection is given by

$$p_{i+4(j-1)} = D_j S_i$$

though there are simplifications that can be made due to redundancies in the scattering matrix as noted in Kaul 2004 and Hayman 2012.

So to optimize the experiment, you need to determine the values of the transmitted Stokes vectors and the receiver diattenuation vectors. The constraints on the polarization states are encompassed by the definitions of those vectors. The exact waveplate angles to achieve those states can be determined afterward. There is no need to write these expanded matrix equations of multiple polarization elements to describe and optimize the measurement.

Also notable, if you vectorize your matrix definitions, you can perform principal component analysis on them to determine the dominant modes in the model study. That also provides a basis for determining the optimal configuration of the system.

---

## Author Response (AR1)

We thank the both reviewers for their constructive comments and suggestions.

We provide below the answers to these comments, along with the corresponding changes in the manuscript.

**Response to reviewer #1**

**Reviewer Comment (RC):**

My biggest concern with this work is that it seems incomplete. After a description of the instrument, the data example is lacking. There is hardly any data, and all of that data is shown as time integrated plots. The integration times are very long (10-15 min -- well in excess of variability in atmospheric structure) and separated considerably in time. This needs some explanation. Also the authors should really be showing time resolved plots which tend to be more revealing, allow the reader to see both the vertical and temporal structure (see, for example, the exceptional plots in figure 8 of Kokhanenko 2020, 10.5194/amt-13-1113-2020).

I don't understand why there is so little data. What is the barrier to running this instrument continuously (a very important question for an instrument paper)? The only thing we are shown is a dust layer with no observable orientation, so we have no assurance that the instrument can observe off diagonal elements. The authors could operate the instrument to observe rain, which has very strong orientation signatures (see Hayman 2014 10.1364/OE.22.016976). That would at least provide some coverage of the measurement space. Demonstration would not be fully complete given the intended application, but it may be asking too much to demand the authors to show polarization properties of oriented dust.

**Reply:**

Although we agree with the reviewer that it would be better to show measurements of rain orientation, we haven't managed to acquire them by now, due to the technical challenges these measurements entail, mainly due to the analog detection of our signals, which are saturated from overlaying clouds and/or the rain. Although this is not impossible to cope, it requires extensive experimentation, which we think it is out of the scope of this paper.

Another issue is that we had to go through repairs for the lasers (once due to laser malfunction and once due to improper operation), which delayed our field measurements.

We decided that in order to avoid confusion, we include a dust-free case to the "First measurements" section, which shows no orientation (as expected). We use these measurements to show that the instrument works as expected and provides "no orientation" flags, for dust-free atmospheres. The measurements used were acquired at viewing angle of 80o off-zenith, to highlight the scanning capabilities of the system. Moreover, we provide the Rayleigh fit of the lidar signals, as a quality standard of our measurements.

We followed the advice of the reviewer and we present our measurements with time-resolved plots.

**RC:**

Given that this is an instrument paper, I would think operability is part of the design and performance criteria. Is this somehow connected to the very small set of observation examples?

**Reply:**

We have included some more information about the operability of the system in Section 2, which we renamed "Overview of the lidar components and operation"

We added the following paragraph in line 68: " Due to the analog operation at 1064 nm, the time range of the measurements is restricted by the dark signal changes, which are mainly affected by the change of the (internal) system temperature. The investigation of the acceptable temperature changes, and corresponding acceptable time ranges during which the dark signal does not change considerably, is a work in progress, with first results to set the acceptable temperature changes to ±2 oC, which require a new dark measurement every 0.5 hour during summertime, or every 2 hours during wintertime. Also, due to the high power of the lasers there is no eye safety classification for the lidar, although the beam is expanded 5 times. This restricts the operation of the system when there are no aircrafts at the airspace of the measurements."

**RC:**

I am somewhat concerned that the authors seem to have decided that oriented dust is a foregone conclusion. The published work on this phenomena appears to be circumstantial (see specific comment about Line 13), so I would recommend the authors adopt a more cautious tone on the subject.

**Reply:**

There has been a revision on this assertion. We keep a more cautious tone throughout the manuscript and we added in the last section (8. Overview and future perspectives) the following: "Currently, the only indication of particle orientation comes from astronomical polarimetry measurements of dichroic extinction, which though cannot provide a strong proof for the phenomenon, due to their small number."

**RC:**

It is notable that there is no discussion of uncertainty in this work. This seems like a pretty important aspect of the instrument design.

**Reply:**

The quantification of the uncertainty is a work in progress, which entails e.g. an extensive investigation of the effects of the analog signal distortions.

In the revised Section 7 ("First measurements") we included a first estimation of the uncertainty level of the measured orientation flags, by providing the standard deviation of the measurements with height which is quantified to be ±2-10% for the averaged signals. Although this is not an optimum quantification of the uncertainty of the measurements (e.g. since it may include real variability), this is a preliminary estimation and more extensive testing and analysis will follow.

Moreover, we included the following in the last section (8. Overview and perspectives): " Moreover, an extended analysis should be performed to characterize the analog signal distortions, which are expected to affect the quality of the signals. Specifically, we will investigate their dependence on temperature and signal strength, and their change with height range and time. A very preliminary indication of their effect is shown in the first measurements presented herein, with a standard deviation of the measured orientation flags of 2–10%."

**RC:**

Line 13: "Dust particles have non-spherical irregular shapes and they have been reported to present preferential orientation (Ulanowski et al., 2007)."

It's worth noting that the analysis presented by Ulanowski is circumstantial. Dichroism from starlight was observed and those authors, lacking another explanation, assert that it must be caused by vertically oriented dust. This is not scientifically rigorous proof of oriented dust. The limits of imagination do not constitute scientific proof. (Remember when, lacking any other explanation, a neutrino traveled faster than the speed of light at CERN?).

 The correct assertion is that dichroism has been observed in starlight when Saharan dust was present and that has led to the hypothesis that Saharan dust could have a preferential orientation. If the conclusions from Ulanowski 2007 et al are already deemed sufficient and correct, why build a lidar to look at this? Clearly there needs to be more, different observations.

**Reply:**

We changed line 18 accordingly: "Specifically, the only indication of dust orientation in the Earth's atmosphere comes from astronomical polarimetry measurements of dichroic extinction during a dust event at the Canary islands (Ulanowski et al., 2007)..."

Moreover, we added in the last section the following: "Currently, the only indication of particle orientation comes from astronomical polarimetry measurements of dichroic extinction, which though cannot provide a strong proof for the phenomenon, due to their small number. "

**RC:**

Line 45:   The authors note that they are using high power lasers.   What is the eye safety classification of the lidar system and does that affect how and when often the instrument can be run?

**Reply:**

In line 45 we added that we use Class 4 lasers.

Moreover, we added in line 68: "Due to the high power of the lasers there is no eye safety classification for the lidar, although the beam is expanded 5 times. This restricts the operation of the system when there are no aircrafts at the airspace of the measurements."

**RC:**

Line 82:   The description of the telescope system does not mention a field stop.   What is the angular acceptance of the receiver?

**Reply:**

The telescope has a field stop with a 2mm diameter. The telescope has a focal length of 1000 mm. This means that the field of view is 2 mrads ((FS_diameter)/focal_length). We changed line 82 accordingly: "The telescopes are of Dall-Kirkham type, with an aperture of 200 mm, focal length of 1000mm (F#5), field stop with diameter of 2mm and a field of view of 2mrads."

**RC:**

Line 86:   "The signals are recorded by two cooled Avalanche PhotoDiodes (APDs) at each detection unit,"

What mode are the APDs operating in?   Analog or geiger?   How are signals acquired and stored?   Are photon counts converted to a histogram, and if so at what time and range resolution?   Are analog signals digitized with A/Ds and at what sample rate and what is the analog bandwidth of the digitizer?

**Reply:**

We added in line 86: "We operate the APDs in Analog mode (not geiger). Signals from APD are pre-amplified and digitized by an 16 bit A/D with a sampling rate of 40 MHz and bandwidth of

DC to 20 MHz. After digitization, the signals are stored as mVolts at the hard disk of the embedded computer."

**RC:**

Line 124: "Moreover, most of the previous works utilize visible light measurements whereas we use near infrared light measurements at 1064 nm, to better probe the larger dust particles (a more detailed discussion is provided in Tsekeri et al. (2021)"

This statement references work that is neither published nor submitted for publication. Please provide some high level explanation for why IR is better for probing dust.

**Reply:**

We use instead the references of the work of Gasteiger and Freudenthaler (2011) and Burton et al. (2016) that support this claim.

**RC:**

Line 141: Eq. 1 is in a strange part of the text. The text immediately above is discussing laser polarization, not the scattering matrix. As a reader, I was confused when I saw the equation.

**Reply:**

We deleted Eq. 1.

**RC:**

Line 162: The transmission term is treated as a scalar (having no polarization effect) in this work, but Ulanowski et al., 2007 specifically measured dichroism in dust extinction. Please state the justification treating the transmission of dust as a scalar.

**Reply:**

We added in line 164: "T(0,R) is simplified to a scalar, since the polarization effect due to the transmission (i.e. dichroism) is deemed to be small (Ulanowski et al., 2007)."

**RC:**

Line 182: Please explain why the elements of $f_{ij}$ and I, Q, U, V are now being treated as vector quantities.

**Reply:**

That was a mistake and it has been corrected.

**RC:**

Line 184 (just below Eq. 8): I think the definition for $g_{ij}$ has the wrong denominator. Shouldn't it be $G_{11}$?

**Reply:**

No, we define g11=G11/F11.

**RC:**

Line 224: "The optical elements are considered to be perfectly aligned with each other in the detection units after telescopes A and B"

Since there is no such thing as perfectly, this raises the question: What are the angle tolerances on the manufacturing and alignment?

**Reply:**

We added in line 86: "The optical elements are well-aligned with each other, considering the high tolerance for misalignment due to the emitting divergence at 0.2 mrad and the field of view of 2 mrad."

And we corrected line 224: "The optical elements are considered to be well-aligned with each other in the detection units after telescopes A and B"

**RC:**

Figure 13: In (c) orientation flag, there appears to be some bias above 1.0 both above and below the dust layer. I would have thought that above the layer, since the orientation is nonlinear, noise could be causing the bias, but below, the noise is quite low. Why is the

orientation flag not equal to 1 when there is plenty of signal? Please provide more discussion on this new retrieved quantity and the observations.

**Reply:**

In the revised version we use new measurements. The biases in the orientation flag are still there at heights <1km. The reason is not understood well yet, but in any case, they are within the standard deviation of the values at 1-1.5km, as stated in the revised Section 7.

**RC:**

I have attached additional comments on the structure of the equations.

**Reply:**

We thank the reviewer for these suggestions. We tried to incorporate them in the main text, so as to simplify the equations.

**Response to reviewer #2**

**RC:**

The main issue for me is that the largest part occupied by the development of the mathematical formulas. Whereas these equations are very important, in my opinion the article that explains them should mainly speak the language of the atmospheric sciences. I would suggest rewriting the text in such a way that the main principles governing the instrument are explained in words to the reader, with the equations relegated to one of the sections not taking up more than 20-30% of the paper. A scientist wishing to skip this section for brevity, should still be able to understand the article. I would also suggest: on one hand, to simplify the math where possible, and on the other hand to expand on the non-mathematical parts.

**Reply:**

We reduced the number of equations in the main text, moving most of them in the Appendices and in the Supplement.

**RC:**

Papers by Daskalopoulou (2020) and Tsekeri (2021) is mentioned, however they have not been submitted yet. I suggest that the main learnings from these papers should be summarised here in the mean time, and/or that a preview should be provided for the reviewers and the colleagues taking part in the interactive discussion.

**Reply:**

The paper by Daskalopoulou et al. (2021) is replaced with the conference paper: Daskalopoulou V., Raptis I. P., Tsekeri A., Amiridis V., Kazadzis S., Ulanowski Z., Metallinos S., Tassis K., and Martin W.: Monitoring dust particle orientation with measurements of sunlight dichroic extinction, 15th COMECAP, conference proceedings, 2021.

The paper by Tsekeri et al. (2021) has been deleted and the main learnings from this work have been added in lines 146-151: " The methodology for defining the optimum measurements includes extensive simulations for different atmospheric scenarios and machine learning tools. Briefly, the backscattered light is simulated for different mixtures of dust particles with realistic sizes and irregular shapes, including cases with random and preferential particle orientation. We investigate a large number of possible polarizations for laser B, and we evaluate their information content based on the performance of the corresponding neural network retrievals that use the simulated lidar measurements to retrieve the oriented dust microphysical properties. This is an ongoing work, with the first results identifying that the emission from laser B should be elliptically-polarized with the angle of the polarization ellipse at 5.6o and degree of linear polarization of 0.866."

**RC:**

The introduction should place the research into context more. At present, the general presentation of the atmospheric science problem on dust orientation is discussed in the first 10 lines, and I believe that the topic deserves more, together with previous observations and to hypotheses on why it is believed to happen (e.g. dust electrification). See e.g. Nicoll et al (Env. Res. Lett 2010), Merrison et al (Plan. Sp. Sci. 2012), van der Does (Sci. Adv. 2018), Toth III (Atmos. Chem. Phys. 2020), Mallios et al (J. Aer. Sci. 2020, 2021). The topic of mineral dust in general could also be introduced before discussing the specific topic, citing a number of articles (easy to locate as there is plenty of literature), and mentioning the main points that need investigation (composition, particle size and shape, transport mechanisms, gaps in the observations, radiative effects, etc.) and the main methods used (in situ, remote sensing, modelling, etc.). The main applications of this research could also be mentioned.

**Reply:**

A brief description of the a) importance of dust for climate and ecosystems, b) the retainment of the large dust particles for longer distances than explained from their gravitational settling, c) the possible explanation due to dust electrification and d) the orientation of dust along the electric field, have been added in line 12.

**RC:**

There are some points which are unclear as well, and I suggest could be more explicitly be clarified, e.g. is the lidar a scanning one? It sounds like yes at the beginning, but later on there is a sentence about not using any moving parts. What is the preferred viewing geometry and why? Is the orientation controlled through a stepped motor, or is it manual?

**Reply:**

The lidar has scanning capabilities. The text in line 128 refers to the moving parts used for the emission or detection of light. In order to clarify we added in line 128: "... without using any moving parts for the emission or the detection of light"

Moreover, in the "First measurements" Section, we show measurements acquired using a viewing angle of 80o off-zenith, in order to emphasize the scanning capabilities of the system.

There is no "preferred" viewing angle, since this depends on the orientation angle of the particles.

The viewing geometry (i.e., zenith and azimuth angles) are controlled manually. We added in line 95: "The positioning of the head at various viewing angles is controlled manually."

**RC:**

Angles are expressed with respect to the horizon, but to the reader it is not fully clear what this means: it seems to make sense perhaps for a horizontal observation but not e.g. for a zenith geometry. I admit that I got lost with the different angles expressed in the article and that it should be made clearer every time what are the two planes between which an angle is measured.

**Reply:**

The horizon is used to describe the x-axis of the "frame coordinate system", based on which we describe all the different geometries of the system. The change to the zenith geometry would require a major rewrite of the parts describing all different geometries of the system.

We agree with the reviewer that it is difficult to follow the discussion about all the different angles. This is the reason we provide a very detailed description of all the angles in Fig. 8.

**RC:**

The hardware set-up of the receiver could probably be better illustrated with a drawing than with Figure 3.

**Reply:**

We moved the drawing of the system (Fig. 7) in the beginning of Section 2.

**RC:**

The function of some units of the hardware (LPC, precipitation sensor, UPS) is leaved implicit. I believe it should be explicit (e.g. "detection of precipitation causes shutdown of the lidar", "UPS can keep the system running for X hours in case of power failure", "the purpose of LPC is XXX", etc.). You also mention shutting down the lasers in case of emergency: what type of emergency and how is it detected?

**Reply:**

We provide more information about the function of the LPC, the precipitation sensor, the UPS and the hardware interlocks:

For the LPC, the precipitation sensor and the hardware interlocks (i.e., external push buttons that shut down the system in case of emergency) we changed lines 113-118 as following: "The lidar system is controlled from the LPC unit. This is an "enhanced" embedded computer with specific I/Os that fits the lidar requirements, providing several ethernet interfaces that make

the controlling (local or remote) of the lidar easy and safe. The LPC controls all lidar sub-components (e.g. the lasers, data acquisition systems), along with any auxiliary equipment used by the lidar system (e.g., the precipitation sensor, temperature and humidity sensors, cameras for the alignment). Additionally, it controls the mechanical rotators of the optical elements used for calibration purposes (Section 4), and it stores the acquired raw measurements. The precipitation sensor (Fig. 6) provides information about precipitation conditions and causes shutdown of the lidar when precipitation is detected. Moreover, several external easy accessible push buttons are connected to the LPC and can be used by the operators to shut down the lasers in case of emergency."

We added the info for the UPS in line 102: "The UPS can provide power to the system for about one hour, in case of power failure. This is enough time for a proper cool down of the lasers and shutting down of the system."

**RC:**

In general the reasons behind the design choices could be given: why two telescopes and not e.g. a single telescope with a more complex optical system behind, allowing the same states of polarisation to be measured? Why does the second laser emit elliptically polarised light and not circular polarised, and how is the optimal polarisation ellipse chosen?

**Reply:**

The two lasers/two telescopes configuration helps in achieving good signal-to-noise-ratio in short measurement times. This was mentioned in the abstract, but it is now included in Section 2 as well, in lines 59-60: "The system uses this "two-laser/two-telescope/four-detector" setup to record eight separate signals with good SNR in short measurement times."

The definition of the polarization of laser B is a work in progress. We clarify this in the manuscript by changing lines 146-151: "The methodology for defining the optimum measurements includes extensive simulations for different atmospheric scenarios and machine learning tools. Briefly, the backscattered light is simulated for different mixtures of dust particles with realistic sizes and irregular shapes, including cases with random and preferential particle orientation. We investigate a large number of possible polarizations for laser B, and we evaluate their information content based on the performance of the corresponding neural network retrievals that use the simulated lidar measurements to retrieve the oriented dust microphysical properties. This is an ongoing work, with the first results identifying that the emission from laser B should be elliptically-polarized with the angle of the polarization ellipse at 5.6o and degree of linear polarization of 0.866."

**RC:**

First measurements are shown very briefly and they show that the system works, but the case study chosen does not allow to highlight particle orientation (the main goal of this new

instrument). I would support Anonymous Referee #1's suggestion that it would be useful to show an example where particle orientation is observed (not necessarily dust if an example has not yet been identified).

**Reply:**

Although we agree with the reviewer that it would be better to show measurements of e.g. rain orientation, we haven't managed to acquire them by now, due to the technical challenges these measurements entail, mainly due to the analog detection of our signals, which are saturated from overlaying clouds and/or the rain. Although this is not impossible to cope, it requires extensive experimentation, which we think it is out of the scope of this paper.

Another issue is that we had to go through repairs for the lasers (once due to laser malfunction and once due to improper operation), which delayed our field measurements.

We decided that in order to avoid confusion, we include a dust-free case to the "First measurements" section, which shows no orientation (as expected). We use these measurements to show that the instrument works as expected and provides "no orientation" flags, for dust-free atmospheres. The measurements used were acquired at viewing angle of 80o off-zenith, to highlight the scanning capabilities of the system. Moreover, we provide the Rayleigh fit of the lidar signals, as a quality standard of our measurements.

**RC:**

Finally, the 1-page long overview and future perspectives section is merely a summary of the article followed by a brief description of future plans. I believe that it would be useful to tie the research more widely to the wider field of research, going back to the main questions raised in the introduction and explaining how you are contributing to answer some of them. This section could be completely rewritten.

**Reply:**

We revised the Section "Overview and future perspectives" accordingly, trying to tie the work presented in the manuscript to the wider field of research.

---

## Author Response (AR2)

We thank the Editor and the Anonymous Referee #1 for their constructive comments and suggestions.

We provide below a point-by-point response to these comments, along with the corresponding changes in the manuscript.

Moreover, some more (minor) changes have been applied in the manuscript, mainly correcting typos and syntax errors, which are not reported here, but they are provided in the manuscript with track changes.

**Reply to the Editor**

**Editor Comment (EC):**

Thank you for your response to the reviewers' comments. As you see, reviewer 1 has provided a second round of responses. The main criticism is that while the lidar is designed to detect dust particle orientation, the paper does not provide observations to show that the instrument can detect particle orientation.

I appreciate that there may be difficulties in obtaining such measurements, and as such I consider the instrument description and measurement data in current form valuable for publication. However, there should be a much clearer communication of the fact that the current work and instrument has not yet detected oriented dust (or other orientated particles), and therefore the instrument has not been tested fully in this objective. This should be included and more explicitly defined in the abstract, main article, and Overview/future perspective section. An explanation of the difficulties in using rain as an example should also be covered in the article (as in the previous responses to reviewers).

**Reply:**

In order to emphasize the fact that our instrument has not yet detected particle orientation, we included the following in the abstract, main article, and Overview/future perspective sections:

Line 12: "The work presented does not include the detection of oriented dust (or other oriented particles), and therefore the instrument has not been tested fully in this objective."

Line 69: "In Section 7 we present the first measurements of the system, acquired during a dust-free case in Athens, Greece (we should note that the instrument has not acquired measurements of oriented particles yet)."

Line 377: "Further work is required towards improving the system performance and fulfill its objective. Firstly, the system has not yet measured dust orientation, or the orientation of other particles (e.g. rain as in Hayman et al. (2012)), thus it is not fully tested in this respect. The detection of the orientation of rain has been tried, but it hasn't been completed, since it entails high technical challenges, mainly due to the analog detection of the signals which are saturated from overlaying clouds and/or the385rain. Although this is not impossible to cope, it requires extensive experimentation, which will be part of our future work."

**EC:**

In addition, please respond to the most recent response from the reviewer, and make appropriate changes to the article.

**Reply:**

We provide a point-by-point response to the reviewer's comments below.

**EC:**

Finally, it seems unclear whether the example presented relates to a dust-free case, or a dust case with no orientation. It seems that the description of the aerosol case has been changed from dusty (first submission) to dust-free (second submission). Please explain the justification for changing this description in your response.

**Reply:**

The presented case is dust-free, as indicated by the dust transport simulations from the WRF-Chem model (no transport of dust particles in Athens and in the Mediterranean Sea region). These results are also supported from close-to-zero VLDR measurements at 532 nm (indicating spherical -no dust- particles), acquired with the PollyXT lidar of NOA at the PANhellenic GEophysical observatory of Antikythera (PANGEA).

In order to clarify this in the text, we have included the following in the manuscript, in line 345: "The absence of dust is supported from the WRF-Chem model simulations, indicating that desert dust has not been advected over the region, as well as from the low values of VLDR at 532 nm, measured with the PollyXT lidar of Antikythera (Baars et al., 2016), indicating spherical particles (not shown here)."

The justification for changing the dusty case (presented in the first submission) to the dustfree case (in the second submission) is provided in the discussion of the second submission: Due to the absence of measurements of dust orientation, in order to avoid confusion, we include a dust-free case to the "First measurements" section, which shows no orientation (as expected). We use these measurements to show that the instrument works as expected and provides "no orientation" flags, for dust-free atmospheres.

**Reply to Reviewer #1**

**Reviewer Comment (RC):**

The authors have made some positive revisions to the manuscript. They have provided additional details about the instrument specifications and added 2D plots of the observations (which look very nice). However, there is not an explanation about how the standard deviation of the derived data products is obtained. Most significantly, I would note that the shared comment from both previous reviewers has not been substantively addressed: There is no demonstrated observation of any oriented particles in this manuscript.

**Reply:**

Regarding the standard deviation of the data products, it is derived as the standard deviation of the values of the orientation flags at a specific range of heights.

We added the clarification in line 354: "...(the standard deviation is calculated as the variation of the values of orientation flags from the full-overlap height up to 1.5 km)."

**RC:**

The standard for what warrants publication of an instrument paper ultimately rests with the editors. I feel this work does not meet my standard largely because I feel an instrument needs to prove out the parameter space of it's measurements. My experience is that an instrument that has not demonstrated its purpose (in this case detecting oriented particles) is probably not done. It is very possible that aspects of the instrument design and operation will change in the process of attempting to measure particle orientation through polarization. If this work is published as it is, that would potentially result in a published design which is not consistent with the system eventually in operation. It does not appear to me that this research effort has yet achieved a milestone that is appropriate for publication.

To be clear, this work shows that the instrument does not detect orientation when there is clear air and in at least one instance of dust where we also assume the particles are not oriented. That helps assure us that it doesn't suffer from excessive false positives and captures true negatives. But how well does the instrument perform when there are oriented particles? That is undemonstrated, but it is the entire novelty of the instrument.

This is not the first lidar designed to measure oriented particles in the atmosphere. It is, as stated in the manuscript, the first lidar (that I am aware of) designed to detect oriented dust. A review of previous publications on oriented particle detection lidar includes example cases (e.g. Kaul et al 2004, Hayman et al 2012). So while I support the eventual publication of this manuscript, I think the authors need to do more work.

**Reply:**

As responded to the EC above, in order to emphasize this limitation of our work, we included the following in the manuscript:

Line 12: "The work presented does not include the detection of oriented dust (or other oriented particles), and therefore the instrument has not been tested fully in this objective."

Line 69: "In Section 7 we present the first measurements of the system, acquired during a dust-free case in Athens, Greece (we should note that the instrument has not acquired measurements of oriented particles yet)."

Line 377: "Further work is required towards improving the system performance and fulfill its objective. Firstly, the system has not yet measured dust orientation, or the orientation of other particles (e.g. rain as in Hayman et al. (2012)), thus it is not fully tested in this respect. The detection of the orientation of rain has been tried, but it hasn't been completed, since it entails high technical challenges, mainly due to the analog detection of the signals which are saturated from overlaying clouds and/or the385rain. Although this is not impossible to cope, it requires extensive experimentation, which will be part of our future work."

**RC:**

I do not think that the request that this instrument is able to detect some oriented particles (not necessarily dust) is unreasonable. As has already been discussed, there are some predictable cases where these observations can be made. If the detectors are not linear in cases of rain, why can't the laser or detection path be attenuated? This should not be a particularly difficult task. In any case, the problems the authors have encountered may be inconvenient but they are not insurmountable.

**Reply:**

We agree with the reviewer that the problems encountered for measuring the orientation of rain are not insurmountable, but they do require a lot of experimentation and time, which we choose to include as future work.

**RC:**

Finally I would make a comment about instrument design that should not be viewed as an argument against publishing this manuscript:

While this system is designed to meet the technical requirements to measure the polarization signatures of oriented dust, I am skeptical that the design meets the practical requirements. If oriented dust is relatively rare, then building an instrument that requires attended operation with high power (non-eye-safe) lasers is going to make detecting such instances quite difficult. Continuous operation is a likely requirement to perform any meaningful study of this phenomenon.

**Reply:**

We thank the reviewer for his/her insight. A continuous-operation instrument would be much more preferable for many reasons, but it is also much costlier and its construction is more challenging. In any case, the design incorporates parts that can be upgraded for continuous and automatic operation in the future. This aspect is not discussed in the manuscript since it is in a very preliminary stage. The system will be applied in a forthcoming experimental campaign in Cape Verde to measure possible particle orientation within the SAL. Intensive measurements will be taken, and the high-power laser operations will be interrupted automatically in case of an aircraft overflight will be detected over the site (through a radar that will operate along with the lidar to apply all eye-safety measures for our operations).

---

## Author Response (AR3)

We thank the Editor and the Anonymous Reviewers for their constructive comments and suggestions. As the Editor noted, we hope this work will lead to further exciting measurements!